# Multicomponent Stress–Strength Model Based on Generalized Progressive Hybrid Censoring Scheme: A Statistical Analysis

**DOI:** 10.3390/e24050619

**Published:** 2022-04-29

**Authors:** Haijing Ma, Zaizai Yan, Junmei Jia

**Affiliations:** College of Science, Inner Mongolia University of Technology, Hohhot 010051, China; 20191000015@imut.edu.cn (H.M.); jjm606@imut.edu.cn (J.J.)

**Keywords:** multicomponent stress–strength model, generalized progressive hybrid censoring, reliability, maximum likelihood estimation, Bayesian estimation

## Abstract

The statistical inference of the reliability and parameters of the stress–strength model has received great attention in the field of reliability analysis. When following the generalized progressive hybrid censoring (GPHC) scheme, it is important to discuss the point estimate and interval estimate of the reliability of the multicomponent stress–strength (MSS) model, in which the stress and the strength variables are derived from different distributions by assuming that stress follows the Chen distribution and that strength follows the Gompertz distribution. In the present study, the Newton–Raphson method was adopted to derive the maximum likelihood estimation (MLE) of the model parameters, and the corresponding asymptotic distribution was adopted to construct the asymptotic confidence interval (ACI). Subsequently, the exact confidence interval (ECI) of the parameters was calculated. A hybrid Markov chain Monte Carlo (MCMC) method was adopted to determine the approximate Bayesian estimation (BE) of the unknown parameters and the high posterior density credible interval (HPDCI). A simulation study with the actual dataset was conducted for the BEs with squared error loss function (SELF) and the MLEs of the model parameters and reliability, comparing the bias and mean squares errors (MSE). In addition, the three interval estimates were compared in terms of the average interval length (AIL) and coverage probability (CP).

## 1. Introduction

The stress–strength model is used extensively in mechanical engineering. The model plays a crucial role in designing and analyzing the reliability of system equipment. In the model, the reliability of a system is described by the relationship between the strength of the system and the stress applied to the system. If the strength of the system is unable to resist the stress applied to the system, the system fails. Therefore, R=P(Y<X) represents the reliability of the system, where *X* denotes the strength and *Y* denotes the stress applied to the system. Birnbaum [1] was the first to propose the Mann–Whitney statistic, following which the formal terms ‘stress’ and ‘strength’ appeared in the report by Church and Harris [2]. Since then, the stress–strength model has been studied extensively in terms of statistics and reliability. Guo and Krishnamoorthy [3] proposed a novel method to analyze the reliability of the stress–strength model using unknown parameters when the stress and strength variables have a normal distribution. Khan and Jan [4] studied the reliability estimation of the model by assuming that the stress and strength variables obeyed the two-parameter Lindley distribution. Kundu and Raqab [5] calculated the estimation for R=P(Y<X), where both *X* and *Y* have generalized Rayleigh distributions. These authors explained the influence of the mixed proportion parameters on the reliability of the model.

In fact, various products have been developed as technology advances, ranging from simple single-component systems to complex multicomponent subsystems. Multi-component-system products are the mainstream in recent years. Therefore, studying the reliability of multicomponent systems in the stress–strength model (MSS) has become the focus of reliability research. At present, the *s*-out-of-*j* system has gained extensive attention in the fields of engineering and precision equipment design; we call it multicomponent system in reliability analysis research. The system comprises *j* components, and the strength of each component $X_{1},X_{2},\ldots ,X_{j}$ is independent and identical. Meanwhile, the system is subjected to a stress *Y*. Such a system works when s(1≤s≤j) or more components operate simultaneously. The series system and parallel system are special cases of a multicomponent system. Currently, greater attention is being paid to the multicomponent system, which is capable of obtaining a high-reliability system using low-reliability components. For instance, the solar power system includes *j* solar panels. The power generation system would supply power only when at least *s* solar panels generate electricity normally. In irrigation techniques in agriculture, if the storage capacity of the reservoir in one month of the year exceeds that in the month of August of the previous year, it is considered that there would be no drought in that year. Therefore, the storage capacity of the reservoir in the month of August of the previous year is regarded as the stress, while the storage capacity of the reservoir between January and June of the next year is regarded as strength. Therefore, the stress–strength model of the 1-out-of-6 system may be used for analyzing the problem. The reliability analysis for the MSS model under exponential distributions was studied by Kunchur and Munoli [6]. Rao [7,8,9], on the other hand, studied the reliability of the MSS model when the random stress and strength variables have generalized exponential distributions, Burr-XII distributions, and two-parameter exponentiated Weibull distributions. Khan [10] and Liu [11] studied a stress–strength model for the *n*-component-standby system. Eryilmaz [12] reported certain conclusions regarding the modeling of multi-state systems. In mechanical engineering and automation, since multicomponent systems involve multiple components, some of the precision components are too expensive and must be tested through censoring life experiments.

Censoring samples may be particularly difficult for statistical analysis work. If the sample is not accounted for and processed, the analytical results would be erroneous. In this context, different statistical techniques are employed to deal with the corresponding censoring methods. Using progressive type-II censored samples and the MSS model, Valiollahi et al. [13], Rezaei et al. [14], and Baratpour [15] analyzed the reliability estimation for the Weibull distribution, generalized Pareto distribution, and proportional hazard models, respectively. Recently, Shi [16,17] analyzed product-life reliability, based on the adaptive progressive type-II hybrid censoring scheme and progressive first-failure censoring. Under hybrid censoring schemes, Asgharzadeh [18] estimated the reliability of the MSS model when *X* and *Y* obey two independent Weibull distributions. Mirjalili et al. [19] discussed the stress–strength reliability for exponential distributions. Furthermore, based on a progressive hybrid censoring scheme, Bai [20] derived the reliability of the dependent stress–strength model using maximum likelihood and bootstrap methods.

Most of the systems stated above assume that the stress and strength variables have the same distribution; accordingly, the characteristics of the stress–strength model are analyzed. However, as stated in [12,20], stress and strength variables are independent and could have different distributions. Therefore, analyzing the reliability of MSS models with different distributions is meaningful work. In terms of the cost, testing duration, and accuracy of the reliability estimates, popular censoring schemes such as hybrid censoring and progressive hybrid censoring offer higher test efficiencies, compared with the traditional type-I and type-II censoring schemes. The GPHC scheme is the extension of progressive hybrid censoring. This scheme improves test efficiency by allowing experimenters to observe a sample with a pre-specified number of failure units prior to the final termination point. However, literature on MSS models based on the censoring scheme is scarce.

In the present study, to reduce the cost and time as much as possible, a stress–strength model of a multicomponent system with wider application and more accurate reliability estimation is obtained. On the one hand, the stress and strength variables were assumed to have different distributions based on the GPHC scheme. On the other hand, the estimated values of the model parameters and reliability were obtained using the mathematical-statistical method. Specifically, the following issues were studied:(1)Comparison between MLE and BE, in terms of the reliability estimation of the MSS model;(2)Influence of the GPHC scheme on the reliability estimation of the MSS model;(3)Theoretical basis for the exact interval of the model parameters and a comparison between the ECI, ACI, and HPDCI, in terms of AIL and CP.

The remaining portion of the paper is organized as follows. In Section 2, the model description and the reliability *R* of the MSS model, determined under the GPHC scheme, are discussed. The derivation of the MLEs, ACIs, ECIs, BEs, and HPDCI for the parameters and the *R* of the MSS model are presented in Section 3. Section 4 presents the numerical experiment based on Monte Carlo simulations and real data analysis. The conclusions are provided in Section 5.

## 2. Model

Gompertz [21] was the first to introduce the Gompertz distribution to fit tumor growth. The probability density function (PDF) and cumulative distribution function (CDF) are given by: (1)$$g\left(x;\alpha ,\eta \right)=\alpha e^{\eta x}e^{-\frac{\alpha }{\eta }\left(e^{\eta x}-1\right)},\;\;x>0$$
and: (2)$$G\left(x;\alpha ,\eta \right)=1-e^{-\frac{\alpha }{\eta }\left(e^{\eta x}-1\right)},\;\;x>0,$$
where η>0 and α>0 are the scale and shape parameters, respectively. In the paper, the Gompertz distribution with scale parameter η>0 and shape parameter α>0 is expressed as G(η,α). Recent research, such as El-Gohary [22], has shown that this distribution is widely used in modeling survival time, human mortality, and actuarial tables.

Chen [23] proposed a novel lifetime distribution with a flexible failure rate function. The PDF and CDF of the distribution are: (3)$$f\left(y;\theta ,\beta \right)=\theta \beta y^{\beta -1}\exp\left\{\theta \left(1-e^{y^{\beta }}\right)+y^{\beta }\right\},\;\;y>0$$
and: (4)$$F\left(y;\theta ,\beta \right)=1-\exp\{\theta \left(1-e^{y^{\beta }}\right)\},\;\;y>0,$$
respectively, where θ>0 and β>0 are shape parameters. We recorded the Chen distribution as Ch(θ,β), where θ and β are shape parameters. If θ=1, this distribution simplifies an exponential power distribution. One could also mention Singh et al. [24] and other references, where a few recent works on the Chen distribution are cited.

Tian [25] proposed a GPHC life test scheme; we consider a life test involving *n* units. We assumed $X_{1},X_{2},\ldots ,X_{n}$ as the life of each unit. Before the life test, the following steps must be taken:(1)Setting the fixed integer c,m to satisfy the relationship c<m<n. This could save costs. Because of the life test, if all units fail, it will cause a certain cost loss;(2)Setting the censoring scheme to $R=(R_{1},R_{2},\ldots ,R_{m})$, with the relationship $R_{1}+R_{2}+\ldots +R_{m}+m=n$. This ensures that the samples we obtain are both failed and non-failed, which increases the richness of the samples and helps to improve the accuracy of statistical inference;(3)Setting time *T*. It is the test time limit and a bounded integer;(4)Calculating $T^{*}=\max\left\{X_{c:m:n},\min\left\{X_{m:m:n},T\right\}\right\}$. As metioned in [25], $T^{*}$ is the end of the experiment. By combining the expression of $T^{*}$ and setting $R=(R_{1},R_{2},\ldots ,R_{m})$, we can obtain rich samples (including failure and non-failure) in a shorter time.

Based on these preparations, the life tests are carried out as follows: the failure time of the first observation is $X_{1:m:n}$, and $R_{1}$ is randomly removed from $n_{1}-1$; the failure time of the second observation $X_{2:m:n}$ is tested, and $R_{2}$ is randomly removed from the survival units, and so on. When the experiment reaches $T^{*}$, the experiment ends, and all remaining test units are removed. Therefore, the observation samples under this censoring scheme and the expression of $T^{*}$ are in the following three cases: $$\begin{array}{cc} & \mathrm{Case}\;\;I:\left\{X_{1:m:n},\ldots ,X_{c:m;n}\right\},\;T<X_{c:m;n}<X_{m:m:n}\\ & \mathrm{Case}\;\;\mathrm{II}:\left\{X_{1:m:n},\ldots ,X_{c:m;n},\ldots ,X_{J:m;n}\right\},\;X_{c:m;n}<T<X_{m:m:n}\\ & \mathrm{Case}\;\;\mathrm{III}:\left\{X_{1:m:n},X_{2:m;n},\ldots ,X_{m:m;n}\right\},\;X_{c:m;n}<X_{m:m:n}<T \end{array}$$
where *J* is the number of failures before *T*, and $X_{J:m:n}<T<X_{J+1:m:n}$. In case III, it corresponds to progressive II censoring scheme. We obtain a flexible *s*-out of -*j* system: the G system. Due to that, the sample size in the GPHC scheme is random. Basing on the scheme, the likelihood function is: (5)$$\begin{array}{c} L\left(\Omega \right)=\prod_{i=1}^{J^{*}}\left[f\left(x_{i:m:n}\right)\cdot {\left(1-F\left(x_{i:m:n}\right)\right)}^{R_{i}}\right]\cdot {\left[1-F\left(T^{*}\right)\right]}^{R_{J^{*}}^{*}} \end{array}$$
where: $$\begin{array}{cc} & \mathrm{Case}I:J^{*}=c,T^{*}=x_{c:m:n},R_{J^{*}}^{*}=R_{c}^{*}=\sum_{i=c+1}^{m}\left(R_{i}+1\right)\\ & \mathrm{Case}\mathrm{II}:J^{*}=J,T^{*}=T,R_{J^{*}}^{*}=R_{J}^{*}=n-\sum_{i=1}^{J}R_{i}-J\\ & \mathrm{Case}\mathrm{III}:J^{*}=m,T^{*}=x_{m:m:n},R_{J^{*}}^{*}=R_{m}^{*}=0 \end{array}$$

Bhattacharyya and Johnson [26] developed the stress–strength model for multicomponent systems. The reliability of a stress–strength model corresponding to multicomponent systems can be written as $P(X_{j-s+1:j}>Y)$, where $X_{1},X_{2},\ldots ,X_{j}$ are the strength of the component from the G(η,α), and *Y* is the common random stress, following Ch(θ,β).

Furthermore, $X_{j-s+1:j}$ is the (j−s+1)-th order statistic of $\left(X_{1},X_{2},\ldots ,X_{j}\right)$. Therefore, the reliability of a multicomponent stress–strength model is given by: (6)Rs,j=P(atleastsofthe(X1,X2,…,Xj)exceedY)=∑i=sjji∫0∞(1−G(y;η,α))i(G(y;η,α))j−idF(y;θ,β)=∑i=sjji∫0∞e−αiη(eηy−1)1−e−αη(eηy−1)j−iθβyβ−1eθ(1−eyβ)+yβdy

According to the binomial theorem: 1−e−αη(eηy−1)j−i=∑k=0j−ij−ik(−1)ke−αη(eηy−1)k

Equation (Equation 6) can be reduced to:(7)Rs,j=∑i=sjji∑k=0j−i(−1)kj−ik∫0∞θβyβ−1exp{−αη(eηy−1)(i+k)+θ(1−eyβ)+yβ}dy

## 3. Estimation of Parameters and Reliability

### 3.1. Maximum Likelihood Estimation

To derive the maximum likelihood estimation (MLE) of $R_{s,j}$, we first obtain the MLE of α,η,β,θ. Suppose *N* identical systems are placed in a life testing experiment, each with *K* components. These components are independently and identically distributed, and their strength comes from G(η,α). Correspondingly, the stress on each system comes from Ch(θ,β). As explained in Section 2, GPHC life tests are performed in two stages to obtain the stress and strength GPHC samples, respectively. The stress censoring samples are acquired by the system for the first time, and the strength censoring samples are acquired twice by each component in the system. Tests were performed on the stress of N systems and the strength of K components in each system. First, we set the censoring scheme, $\left\{n_{2},T_{2},c_{2},m_{2},S_{1},S_{2},\ldots ,S_{m_{2}}\right\}$, and calculated $B^{*}=\max\left\{Y_{c_{2}:m_{2}:n_{2}},\min\left\{Y_{m_{2}:m_{2}:n_{2}},T_{2}\right\}\right\}$ to obtain the observed sample of stress. During the life testing experiment, when the first system fails, its corresponding stress is denoted as $Y_{1}$, $S_{1}$ systems are removed from the remaining unfailed systems, and so on. When the time reaches $B^{*}$, the test ends, and all remain systems are removed. We recorded the the stress observation sample as $Y_{1},Y_{2},\ldots ,Y_{M^{*}}$ from Ch(θ,β). Secondly, for the above $M^{*}$ systems, under censoring scheme $\left\{n_{1},T_{1},c_{1},m_{1},R_{1},R_{2},\ldots ,R_{m_{1}}\right\}$, we obtained $T^{*}=\max\left\{X_{ic_{1}:m_{1}:n_{1}},\min\left\{X_{im_{1}:m_{1}:n_{1}},T_{1}\right\}\right\}$. Analogously, when the first component fails, its strength is denoted as $X_{i1}$, then $R_{1}$ systems are removed from the remaining unfailed component, and so on. When the time reaches $T^{*}$, the test is terminated, and all remain components are removed. So, the observed samples of strength are $X_{i1},X_{i2},\ldots ,X_{iJ^{*}},\;i=1,2,\ldots ,M^{*}$ from G(η,α). In this case, the sample can be constructed as follows: $$\left(\begin{array}{cccc} X_{11} & X_{12} & \ldots & X_{1J^{*}}\\ \vdots & \vdots & \vdots & \vdots \\ X_{M^{*}1} & X_{M^{*}2} & \ldots & X_{M^{*}J^{*}} \end{array}\right)\left(\begin{array}{c} Y_{1}\\ \vdots \\ Y_{M^{*}} \end{array}\right)$$

This is a MSS system, and the likelihood function is given by: $$\begin{array}{cc} L(\alpha ,\eta ,\theta ,\beta |x,y)= & \;\;\prod_{i=1}^{M^{*}}\prod_{t=1}^{J^{*}}g\left(x_{it}\right){\left(1\;-\;G\left(x_{it}\right)\right)}^{R_{t}}{\left[1\;-\;G\left(T^{*}\right)\right]}^{R_{J^{*}}^{*}}f\left(y_{i}\right){\left(1\;-\;F\left(y_{i}\right)\right)}^{S_{i}}{\left[1\;-\;F\left(B^{*}\right)\right]}^{S_{M^{*}}^{*}}\\ = & \;\;\;\;\;\;\;\;\;\;\;\;\;\;\;\;\;\;\;\prod_{i=1}^{M^{*}}\prod_{t=1}^{J^{*}}\alpha \theta \beta y_{i}^{\beta -1}\exp\left[\eta x_{it}-\frac{\alpha }{\eta }\left(e^{\eta x_{it}}-1\right)\left(R_{t}+1\right)-\frac{\alpha }{\eta }\left(e^{\eta T^{*}}-1\right)R_{J^{*}}^{*}\right]\\ & \;\;\;\;\;\;\;\;\;\;\;\;\;\;\;\;\;\;\;\;\;\;\;\;\;\;\;\;\;\;\;\;\;\;\;\;\;\;\;\;\;\;\;\;\;\;\;\;\;\;\times \exp\left\{\theta \left(1-e^{y_{i}^{\beta }}\right)\left(S_{i}+1\right)+y_{i}^{\beta }\right\}\exp\left(\theta S_{M^{*}}^{*}\left(1-e^{{B^{*}}^{\beta }}\right)\right) \end{array}$$
where $x_{it:m_{1}:n_{1}}$ and $y_{i:m_{2}:n_{2}}$ are the observed samples of strength and stress under the GPHC scheme and MSS model, and are expressed as $x_{it}$ and $y_{i}$. So, the log-likelihood function is: $$\begin{array}{ccc} l(\alpha ,\eta ,\theta ,\beta |x,y) & = & \sum_{i=1}^{M^{*}}\{\sum_{t=1}^{J^{*}}\log\left(\alpha \theta \beta \right)+\left(\beta -1\right)\log\left(y_{i}\right)+\eta x_{it}-\frac{\alpha }{\eta }\left(e^{\eta x_{it}}-1\right)\left(R_{t}+1\right)\\ & & -\frac{\alpha }{\eta }\left(e^{\eta T^{*}}-1\right)R_{J^{*}}^{*}+\theta \left(1-e^{y_{i}^{\beta }}\right)\left(S_{i}+1\right)+y_{i}^{\beta }\}+\theta S_{M^{*}}^{*}\left(1-e^{{B^{*}}^{\beta }}\right) \end{array}$$

Thus, the likelihood equations are: (8)$$\begin{array}{cc} \; & \frac{\partial l}{\partial \alpha }=\sum_{i=1}^{M^{*}}\sum_{t=1}^{J^{*}}\frac{1}{\alpha }-\frac{\left(R_{t}+1\right)\left(e^{\eta x_{it}}-1\right)}{\eta }-\frac{R_{J^{*}}^{*}\left(e^{\eta T^{*}}-1\right)}{\eta }=0\\ \; & \frac{\partial l}{\partial \eta }=\sum_{i=1}^{M^{*}}\sum_{t=1}^{J^{*}}x_{it}-\left(R_{t}+1\right)\left[{\frac{\alpha }{\eta }}^{2}\left(1-e^{\eta x_{it}}\right)+\alpha e^{\eta x_{it}}\right]-R_{J^{*}}^{*}\left[{\frac{\alpha }{\eta }}^{2}\left(1-e^{\eta T^{*}}\right)+\alpha e^{\eta T^{*}}\right]=0\\ \; & \frac{\partial l}{\partial \theta }=\sum_{i=1}^{M^{*}}\frac{1}{\theta }+\left(1-e^{y_{i}^{\beta }}\right)\left(S_{i}+1\right)+y_{i}^{\beta }+\left(1-e^{{B^{*}}^{\beta }}\right)S_{M^{*}}^{*}=0\\ \; & \frac{\partial l}{\partial \beta }=\sum_{i=1}^{M^{*}}\frac{1}{\beta }+\log y_{i}-\theta \left(S_{i}+1\right)y_{i}e^{y_{i}^{\beta }}+y_{i}^{\beta }\log y_{i}-\theta S_{M^{*}}^{*}B^{*}e^{{M^{*}}^{\beta }}=0 \end{array}$$

Since Equation (Equation 8) is complicated, there is no explicit solution. We need to use the Newton–Raphson method to calculate the MLEs of α,η,θ,β. Their maximum likelihood estimates are recorded as $\hat{\alpha },\hat{\eta },\hat{\theta }$, and $\hat{\beta }$. The MLE of $R_{s,j}$ is recorded as ${\hat{R}}_{s,j}$, and its mathematical expression is: (9)R^s,j=∑i=sjji∑k=0j−i(−1)kj−ik∫0∞θ^β^yβ^−1exp{−α^η^(eη^y−1)(i+k)+θ^(1−eyβ^)+yβ^}dy

Further, the variance–covariance matrix is expressed as: $$\left[\begin{array}{cccc} Var(\hat{\alpha }) & Cov(\hat{\alpha },\hat{\eta }) & Cov(\hat{\alpha },\hat{\theta }) & Cov(\hat{\alpha },\hat{\beta })\\ \; & Var(\hat{\eta }) & Cov(\hat{\eta },\hat{\theta }) & Cov(\hat{\theta },\hat{\beta })\\ \; & \; & Var(\hat{\theta }) & Cov(\hat{\theta },\hat{\beta })\\ \; & \; & & Var(\hat{\beta }) \end{array}\right]$$
which was obtained by inverting the observed Fisher information matrix I(α,η,θ,β) at $\hat{\alpha },\hat{\eta },\hat{\theta },\hat{\beta }$, where: (10)$$I=-\left[\begin{array}{cccc} \frac{\partial^{2}l}{\partial \alpha^{2}} & \frac{\partial^{2}l}{\partial \alpha \partial \eta } & \frac{\partial^{2}l}{\partial \alpha \partial \theta } & \frac{\partial^{2}l}{\partial \mu \partial \beta }\\ \frac{\partial^{2}l}{\partial \alpha \partial \eta } & \frac{\partial^{2}l}{\partial \eta^{2}} & \frac{\partial^{2}l}{\partial \eta \partial \theta } & \frac{\partial^{2}l}{\partial \eta \partial \beta }\\ \frac{\partial^{2}l}{\partial \alpha \partial \theta } & \frac{\partial^{2}l}{\partial \alpha \partial \eta } & \frac{\partial^{2}l}{\partial \theta^{2}} & \frac{\partial^{2}l}{\partial \theta \partial \beta }\\ \frac{\partial^{2}l}{\partial \alpha \partial \beta } & \frac{\partial^{2}l}{\partial \eta \partial \beta } & \frac{\partial^{2}l}{\partial \theta \partial \beta } & \frac{\partial^{2}l}{\partial \beta^{2}} \end{array}\right]$$

The 100(1−τ)% ACI of the parameter Ω is given by: $$\begin{array}{c} \left(\hat{\Omega }-z_{\tau /2}\sqrt{Var(\hat{\Omega })},\hat{\Omega }+z_{\tau /2}\sqrt{Var(\hat{\Omega })}\right) \end{array}$$
where Ω = α, η, θ, or β, and $z_{\tau /2}$ is the τ/2-th upper percentile of the standard normal distribution.

To obtain the ACI of the reliability $R_{s,j}$ of the MSS model, we have two lemmas: As $M^{*}\to \infty$, we have:

**Lemma** **1.**

$$\begin{array}{c} \sqrt{M^{*}}\left(\left(\hat{\alpha }-\alpha \right),\left(\hat{\eta }-\eta \right),\left(\hat{\theta }-\theta \right),\left(\hat{\beta }-\beta \right)\right)\to N\left(0,C^{-1}\left(\alpha ,\eta ,\theta ,\beta \right)\right) \end{array}$$

*where $C^{-1}\left(\alpha ,\eta ,\theta ,\beta \right)=\frac{1}{M^{*}}I$.*


**Corollary** **1.**
*Let ϵ=(α,η,θ,β), and denote $\varphi \left(\epsilon \right)=R_{s,j}$ as $M^{*}\to \infty$; then:*

$$\sqrt{M^{*}}\left({\hat{R}}_{s,j}-R\right)\to N\left(0,B^{*}\right)$$

*where $B^{*}=b^{T}C^{-1}b$, $b={\left(\frac{\partial \varphi (\epsilon )}{\partial \alpha },\frac{\partial \varphi (\epsilon )}{\partial \eta },\frac{\partial \varphi (\epsilon )}{\partial \theta },\frac{\partial \varphi (\epsilon )}{\partial \beta }\right)}^{T}$, its mathematical expression is in the Appendix A.*


**Proof** **of** **Corollary** **1.**Using the delta method [27] and the Taylor series expansion, $\varphi (\hat{\epsilon })$ can be expressed as:
$$\begin{array}{ccc} \varphi (\hat{\epsilon }) & = & \varphi \left(\epsilon \right)+{\left[\varphi {}^{\prime }\left(\epsilon \right)\right]}^{T}\left(\hat{\epsilon }-\epsilon \right)+{\left(\hat{\epsilon }-\epsilon \right)}^{T}\varphi^{{}^{\prime \prime }}\left(\epsilon^{*}\right)\left(\hat{\epsilon }-\epsilon \right)/2\\ & \approx & \varphi \left(\epsilon \right)+{\left[\varphi {}^{\prime }\left(\epsilon \right)\right]}^{T}\left(\hat{\epsilon }-\epsilon \right) \end{array}$$
where $\varphi {}^{\prime }\left(\epsilon \right)$ and $\varphi^{{}^{\prime \prime }}\left(\epsilon \right)$ are matrices of the first and second partial derivatives, and $\epsilon^{*}$ is some value between $\hat{\epsilon }$ and ϵ. So:
$$\begin{array}{c} \sqrt{M^{*}}\left[\varphi \left(\hat{\epsilon }\right)-\varphi (\epsilon \right)]\approx \sqrt{M^{*}}{\left[\varphi {}^{\prime }\left(\epsilon \right)\right]}^{T}\left(\hat{\epsilon }-\epsilon \right). \end{array}$$Using Lemma 1, $M^{*}\to \infty$; then, $\hat{\epsilon }\to \epsilon$, and $\varphi \left(\hat{\epsilon }\right)\to \varphi \left(\epsilon \right)$. The variance of φ(ϵ) is:
$$\begin{array}{ccc} Var(\varphi \left(\hat{\epsilon }\right)) & \approx & Var(\varphi \left(\epsilon \right)+{\left[\varphi {}^{\prime }\left(\epsilon \right)\right]}^{T}\hat{\epsilon }-{\left[\varphi {}^{\prime }\left(\epsilon \right)\right]}^{T}\epsilon )\\ & = & Var({\left[\varphi {}^{\prime }\left(\epsilon \right)\right]}^{T}\hat{\epsilon })\\ & = & {\left[\varphi {}^{\prime }\left(\epsilon \right)\right]}^{T}Var\left(\hat{\epsilon }\right)\varphi {}^{\prime }\left(\epsilon \right)\\ & = & {\left[\varphi {}^{\prime }\left(\epsilon \right)\right]}^{T}\left(C^{-1}/M^{*}\right)\varphi {}^{\prime }\left(\epsilon \right) \end{array}$$
According to the central limit theorem, $\varphi \left(\hat{\epsilon }\right)\to N\left(\varphi \left(\epsilon \right),{\left[\varphi {}^{\prime }\left(\epsilon \right)\right]}^{T}\left(C^{-1}/M^{*}\right)\varphi {}^{\prime }\left(\epsilon \right)\right)$. □

The ACI of $R_{s,j}$ can be obtained by replacing α,η,θ,β in $B^{*}$ with their corresponding values, $\hat{\alpha },\hat{\eta },\hat{\theta },\hat{\beta }$. Thus, the 100(1−τ)% ACI for $R_{s,j}$ is: $$\begin{array}{c} \left[{\hat{R}}_{s,j}-z_{\tau /2}\sqrt{\hat{B^{*}}/M^{*}},{\hat{R}}_{s,j}+z_{\tau /2}\sqrt{\hat{B^{*}}/M^{*}}\;\;\right] \end{array}$$

### 3.2. Exact Interval Estimation

Based on the assumption in Section 3.1, let $V_{i:m_{2}:n_{2}}=\theta \left(e^{Y_{i:m_{2}:n_{2}}^{\beta }}-1\right)$: $$\begin{array}{c} \left\{\begin{array}{c} V_{1}^{*}=n_{2}V_{1:m_{2}:n_{2}}\\ V_{2}^{*}=\left(n_{2}-S_{1}-1\right)\left(V_{2:m_{2}:n_{2}}-V_{1:m_{2}:n_{2}}\right)\\ \vdots \\ V_{M^{*}}^{*}=\left(n-S_{1}-S_{2}-\ldots -S_{M^{*}-1}^{*}-M^{*}+1\right)\left(V_{M^{*}:m_{2}:n_{2}}-V_{M^{*}-1:m_{2}:n_{2}}\right). \end{array}\right. \end{array}$$Then, the generalized spacings $V_{1}^{*},V_{2}^{*},\ldots ,V_{M^{*}}^{*}$ are independent and identically distributed as an exponential distribution with a mean of 1. Let: $$\begin{array}{c} C=2V_{1}^{*}=2n_{2}V_{1:m_{2}:n_{2}} \end{array}$$
and: $$\begin{array}{c} D=2\sum_{j=2}^{M^{*}}V_{j}^{*}=2\sum_{j=1}^{M^{*}}\left(S_{j}+1\right)\left(V_{j:m_{2}:n_{2}}-V_{1:m_{2}:n_{2}}\right). \end{array}$$

Then, *C* obeys a $\chi^{2}$ distribution with two degrees of freedom and is expressed as $\chi^{2}$(2). *D* follows a $\chi^{2}$($2M^{*}-2$) distribution, and the *C* and *D* are independent. Then, we define: (11)$$\begin{array}{ccc} B_{1} & = & \frac{D}{(M^{*}-1)C}=\frac{\sum_{j=1}^{M^{*}}\left(S_{j}+1\right)\left(V_{j:m_{2}:n_{2}}-V_{1:m_{2}:n_{2}}\right)}{n_{2}\left(M^{*}-1\right)V_{1:m_{2}:n_{2}}}\\ & & =\frac{1}{n_{2}\left(M^{*}-1\right)}\sum_{j=1}^{M^{*}}\left(S_{j}+1\right)\frac{e^{y_{j:m_{2}:n_{2}}^{\beta }}-1}{e^{y_{1:m_{2}:n_{2}}^{\beta }}-1}-\frac{1}{\left(M^{*}-1\right)} \end{array}$$
and: (12)$$\begin{array}{c} B_{2}=C+D=2\sum_{i=1}^{M^{*}}\left(S_{i}+1\right)V_{i:m_{2}:n_{2}}. \end{array}$$

It can be easily found that for $B_{1}∼F\left(2M^{*}-2,2\right),B_{2}∼\chi^{2}\left(2M^{*}\right)$, $B_{1}$ and $B_{2}$ are independent.

**Lemma** **2.**
*For any $0<b_{1}<b_{2}$, $q\left(\beta \right)=\frac{e^{b_{2}^{\beta }}-1}{e^{b_{1}^{\beta }}-1}$ is a strictly increasing function for β≠0.*


**Lemma** **3.**
*Let:*

$$B_{1}\left(\beta \right)=\frac{1}{n_{2}\left(M^{*}-1\right)}\sum_{j=1}^{M^{*}}\left(S_{j}+1\right)\frac{e^{y_{j:m_{2}:n_{2}}^{\beta }}-1}{e^{y_{1:m_{2}:n_{2}}^{\beta }}-1}-\frac{1}{\left(M^{*}-1\right)}.$$

*Then, $B_{1}\left(\beta \right)$ is strictly increasing in β for any β≠0.*


**Corollary** **2.**
*If $w\geq \frac{\sum_{j=1}^{M^{*}}\left(S_{j}+1\right)\log y_{j:m_{2}:n_{2}}-n_{2}\log y_{1:m_{2}:n_{2}}}{n_{2}\left(M^{*}-1\right)\log y_{1:m_{2}:n_{2}}}$, the equation $B_{1}\left(\beta \right)=w$ has a unique solution for any β≠0.*


**Proof** **of** **Corollary** **2.**By Lemma 3, it is easy to show that $B_{1}\left(\beta \right)$ is a strictly increasing function of β. Moreover,$\lim_{\beta \to -\infty }B_{1}\left(\beta \right)=0,\lim_{\beta \to +\infty }B_{1}\left(\beta \right)=\infty$, and:
$$\begin{array}{c} \lim_{\beta \to 0}B_{1}\left(\beta \right)=\frac{\sum_{j=1}^{M^{*}}\left(S_{j}+1\right)\log y_{j:m_{2}:n_{2}}-n_{2}\log y_{1:m_{2}:n_{2}}}{n_{2}\left(M^{*}-1\right)\log y_{1:m_{2}:n_{2}}} \end{array}$$
Therefore, if:
$$\begin{array}{c} w\geq \frac{\sum_{j=1}^{M^{*}}\left(S_{j}+1\right)\log y_{j:m_{2}:n_{2}}-n_{2}\log y_{1:m_{2}:n_{2}}}{n_{2}\left(M^{*}-1\right)\log y_{1:m_{2}:n_{2}}}, \end{array}$$
then $B_{1}\left(\beta \right)=w$ has a unique solution for any β≠0. □

**Theorem** **1.**
*Suppose that $Y_{j},j=1,2,\ldots ,M^{*}$ are GPHC samples from a sample of size $n_{2}$ from Chen (θ,β), with censoring scheme $n_{2},T_{2},c_{2},m_{2},S_{1},S_{2},\ldots ,S_{m_{2}}$. Then, the 100(1−τ)% ECI for β is:*

$$\begin{array}{c} \left(\varpi \left(Y_{1},Y_{2},\ldots ,Y_{M^{*}},F_{1-\frac{\tau }{2}}\left(2M^{*}-2,2\right)\right),\varpi (Y_{1},Y_{2},\ldots ,Y_{M^{*}},F_{\frac{\tau }{2}}\left(2M^{*}-2,2\right)\right) \end{array}$$

*where 0<τ<1, and $\varpi (Y_{1},Y_{2},\ldots ,Y_{M^{*}},w)$ is the solution of β for the equation:*

$$\begin{array}{c} \frac{\sum_{j=1}^{M^{*}}\left(S_{j}+1\right)e^{Y_{i}^{\beta }}-n_{2}e^{Y_{1}^{\beta }}}{n_{2}\left(M^{*}-1\right)\left(e^{Y_{1}^{\beta }}-1\right)}=w \end{array}$$



**Proof** **of** **Theorem** **1.**$$\begin{array}{c} B_{1}=\frac{\sum_{j=1}^{M^{*}}\left(S_{j}+1\right)\left(V_{j:m_{2}:n_{2}}-V_{1:m_{2}:n_{2}}\right)}{n_{2}\left(M^{*}-1\right)V_{1:m_{2}:n_{2}}}=\frac{\sum_{j=1}^{M^{*}}\left(S_{j}+1\right)V_{j:m_{2}:n_{2}}-n_{2}V_{1:m_{2}:n_{2}}}{n_{2}\left(M^{*}-1\right)V_{1:m_{2}:n_{2}}}\\ =\frac{\sum_{j=1}^{M^{*}}\left(S_{j}+1\right)\left(e^{Y_{i}^{\beta }}-1\right)-n_{2}\left(e^{Y_{1}^{\beta }}-1\right)}{n_{2}\left(M^{*}-1\right)\left(e^{Y_{1}^{\beta }}-1\right)} \end{array}$$
obeys a nF distribution with $2M^{*}-2$ and two degrees of freedom. Thus, for 0<τ<1:
$$\begin{array}{cc} & 1-\tau =P(F_{1-\frac{\tau }{2}}\left(2M^{*}-2,2\right)<\frac{\sum_{j=1}^{M^{*}}\left(S_{j}+1\right)\left(e^{Y_{i}^{\beta }}-1\right)-n_{2}\left(e^{Y_{1}^{\beta }}-1\right)}{n_{2}\left(M^{*}-1\right)\left(e^{Y_{1}^{\beta }}-1\right)}<F_{\frac{\tau }{2}}\left(2M^{*}-2,2\right))\\ & =P(F_{1-\frac{\tau }{2}}\left(2M^{*}-2,2\right)<\frac{\sum_{j=1}^{M^{*}}\left(S_{j}+1\right)e^{Y_{i}^{\beta }}-n_{2}e^{Y_{1}^{\beta }}}{n_{2}\left(M^{*}-1\right)\left(e^{Y_{1}^{\beta }}-1\right)}<F_{\frac{\tau }{2}}\left(2M^{*}-2,2\right))\\ & =P\left(\varpi \left(Y_{1},Y_{2},\ldots ,Y_{M^{*}},F_{1-\frac{\tau }{2}}\left(2M^{*}-2,2\right)\right)<\beta <\varpi (Y_{1},Y_{2},\ldots ,Y_{M^{*}},F_{\frac{\tau }{2}}\left(2M^{*}-2,2\right)\right) \end{array}$$ □

**Theorem** **2.**
*Based on the assumption of Theorem 1, a 100(1−τ)% joint ECI for β and θ is determined by the following inequalities:*

(13)
$$\begin{array}{c} \left\{\begin{array}{c} \varpi \left(Y_{1},Y_{2},\ldots ,Y_{M}^{*},F_{\tau_{1}}\left(2M^{*}-2,2\right)\right)<\beta <\varpi \left(Y_{1},Y_{2},\ldots ,Y_{M}^{*},F_{\tau_{2}}\left(2M^{*}-2,2\right)\right)\\ \frac{\chi_{\tau_{1}}^{2}\left(2M^{*}\right)}{2\sum_{i=1}^{M^{*}}\left(S_{i}+1\right)\left(e^{Y_{i:m_{2}:n_{2}}^{\beta }}-1\right)}<\theta <\frac{\chi_{\tau_{2}}^{2}\left(2M^{*}\right)}{2\sum_{i=1}^{M^{*}}\left(S_{i}+1\right)\left(e^{Y_{i:m_{2}:n_{2}}^{\beta }}-1\right)} \end{array}\right. \end{array}$$

*where 0<τ<1, $\tau_{1}=\frac{1+\sqrt{1-\tau }}{2}$ and $\tau_{2}=\frac{1-\sqrt{1-\tau }}{2}$, and $\varpi (Y_{1},Y_{2},\ldots ,Y_{M^{*}},w)$ is the solution of β for the equation:*

$$\begin{array}{c} \frac{\sum_{j=1}^{M^{*}}\left(S_{j}+1\right)e^{Y_{i}^{\beta }}-n_{2}e^{Y_{1}^{\beta }}}{n_{2}\left(M^{*}-1\right)\left(e^{Y_{1}^{\beta }}-1\right)}=w. \end{array}$$



**Proof** **of** **Theorem** **2.**From Equation (Equation 12), we know that:
$$\begin{array}{c} B_{2}=2\sum_{j=1}^{M^{*}}\left(S_{j}+1\right)V_{j}=2\sum_{j=1}^{M^{*}}\theta \left(S_{j}+1\right)\left(e^{Y_{i:m_{2}:n_{2}}^{\beta }}-1\right) \end{array}$$
follows $\chi^{2}$($2M^{*}$), and that it is independent of $B_{1}$. Then, for 0<τ<1:
$$\begin{array}{ccc} 1\;-\;\tau & = & P\left(F_{\tau_{1}}\left(2M^{*}-2,2\right)<B_{1}\;<\;F_{\tau_{2}}\left(2M^{*}-2,2\right)\right)\;\times \;P\left(\chi_{\tau_{1}}^{2}\left(2M^{*}\right)B_{2}\;<\;\chi_{\tau_{2}}^{2}\left(2M^{*}\right)\right)\\ & = & P\;\left(\varpi \left(Y_{1},Y_{2},\ldots \;,Y_{M^{*}},F_{\tau_{1}}\left(2M^{*}\;\;-\;2,2\right)\right)\;\;<\;\;\beta \;\;<\;\;\varpi \left(Y_{1},Y_{2},\ldots \;,Y_{M^{*}},F_{\tau_{2}}\left(2M^{*}\;\;-\;2,2\right)\;\right)\right)\\ & \times & P\left(\chi_{\tau_{1}}^{2}\left(2M^{*}\right)\;<\;2\sum_{j=1}^{M^{*}}\theta \left(S_{j}\;+\;1\right)\left(e^{Y_{i:m_{2}:n_{2}}^{\beta }}-1\right)\;<\;\chi_{\tau_{2}}^{2}\left(2M^{*}\right)\right)\\ & \;\;\;\;=\;\;\;\; & P\;(\;\varpi \left(Y_{1},Y_{2},\ldots \;,Y_{M^{*}},F_{\tau_{1}}\left(2M^{*}\;\;-\;2,2\right)\right)\;\;<\;\;\beta \;\;<\;\;\varpi \left(Y_{1},Y_{2},\ldots \;,Y_{M^{*}},F_{\tau_{2}}\left(2M^{*}\;\;-\;2,2\right)\;\right),\\ & & \frac{\chi_{\tau_{1}}^{2}\left(2M^{*}\right)}{2\sum_{i=1}^{M^{*}}\left(S_{i}+1\right)\left(e^{Y_{i:m_{2}:n_{2}}^{\beta }}-1\right)}\;<\;\theta <\frac{\chi_{\tau_{2}}^{2}\left(2M^{*}\right)}{2\sum_{i=1}^{M^{*}}\left(S_{i}+1\right)\left(e^{Y_{i:m_{2}:n_{2}}^{\beta }}-1\right)}) \end{array}$$□

Let $U_{it}=\frac{\alpha }{\eta }\left(e^{\eta x_{it}}-1\right)$, $t=1,2,\ldots ,J^{*}$. It can be seen that $U_{i1}<U_{i2}<\ldots <U_{iJ^{*}}$, $i=1,2,\ldots ,M^{*}$ are the GPHC samples from the exponential distribution with a mean of 1. Considering the following transformation: $$\begin{array}{c} \left\{\begin{array}{cc} U_{i1}^{*}=n_{1}U_{i1}\\ U_{i2}^{*}=\left(n_{1}-R_{1}-1\right)\left(U_{i2}-U_{i1}\right)\\ \;\;\;\;\;\;\;\;\;\;\;\;\ldots \\ U_{iJ^{*}}^{*}=\left(n_{1}-R_{1}-R_{2}-\ldots -R_{J^{*}-1}-J^{*}+1\right)\left(U_{J^{*}}-U_{J^{*}-1}\right), & \end{array}\right. \end{array}$$
the generalized spacings $U_{i1}^{*},U_{i2}^{*},\ldots ,U_{iJ^{*}}^{*}$, and $i=1,2,\ldots ,M^{*}$ are independent and identical exponential distributions with a mean of 1. Let: $$\begin{array}{c} Z=2\sum_{i=1}^{M^{*}}U_{i1}^{*} \end{array}$$
and: $$\begin{array}{c} Q=2\sum_{i=1}^{M^{*}}\sum_{t=2}^{J^{*}}U_{ij}^{*}=2\sum_{i=1}^{M^{*}}\sum_{t=1}^{J^{*}}\left(R_{t}+1\right)\left(U_{it}-U_{i1}\right); \end{array}$$
then, the random variable *Z* has a $\chi^{2}\left(2M^{*}\right)$ distribution, *Q* has a $\chi^{2}\left(2M^{*}\left(J^{*}-1\right)\right)$ distribution, and the *Z* and *Q* are independent. We define: (14)$$\begin{array}{cc} T_{1}= & \frac{Q}{(J^{*}-1)Z}=\frac{\sum_{i=1}^{M^{*}}\sum_{t=1}^{J^{*}}\left(R_{t}+1\right)\left(U_{it}-U_{i1}\right)}{\left(J^{*}-1\right)\sum_{i=1}^{M^{*}}U_{i1}^{*}}\\ \; & =\frac{\sum_{i=1}^{M^{*}}\sum_{t=1}^{J^{*}}\left(R_{t}+1\right)\left(e^{\eta x_{it}}-1\right)}{n_{1}\left(J^{*}-1\right)\sum_{i=1}^{M^{*}}\left(e^{\eta x_{i1}}-1\right)}-\frac{1}{J^{*}-1} \end{array}$$
and: (15)$$T_{2}=Z+Q=2\sum_{i=1}^{M^{*}}\sum_{t=1}^{J^{*}}\left(R_{t}+1\right)U_{it}$$

So, $T_{1}∼F\left(2M^{*}\left(J^{*}-1\right),2M^{*}\right)$, $T_{2}∼\chi^{2}\left(2M^{*}J^{*}\right)$, and $T_{1}$ and $T_{2}$ are independent. In order to obtain the joint ECI for η and α, we need the following lemmas.

**Lemma** **4.**
*For any $0<a_{1}<a_{2}$, $g^{*}\left(\eta \right)=\frac{e^{\eta a_{2}}-1}{e^{\eta a_{1}}-1}$ is a strict increasing function of η for any η≠0.*


**Lemma** **5.**
*Let:*

$$\begin{array}{c} T_{1}\left(\eta \right)=\frac{1}{n_{1}}\sum_{i=1}^{M^{*}}\sum_{t=1}^{J^{*}}\left(R_{t}+1\right)\frac{e^{\eta x_{it}}-1}{e^{\eta x_{i1}}-1}-\frac{1}{J^{*}-1}. \end{array}$$

*Then, $T_{1}\left(\eta \right)$ is strictly increasing in η for any η≠0.*


**Corollary** **3.**
*If:*

$$\begin{array}{c} q^{*}\geq \frac{1}{n_{1}}\sum_{i=1}^{M^{*}}\sum_{t=1}^{J^{*}}\left(R_{t}+1\right)\frac{x_{it}}{x_{i1}}-\frac{1}{J^{*}-1}, \end{array}$$

*then the equation $T_{1}\left(\eta \right)=q^{*}$ has a unique solution for any η≠0.*


**Proof** **of** **Corollary** **3.**By Lemma 5, it is easy to show that $T_{1}\left(\eta \right)$ is a strict increasing function of η. Moreover, $\lim_{\eta \to -\infty }T_{1}\left(\eta \right)=0,\lim_{\eta \to +\infty }T_{1}\left(\eta \right)=\infty$, and:
(16)$$\lim_{\eta \to -0}T_{1}\left(\eta \right)=\frac{1}{n_{1}}\sum_{i=1}^{M^{*}}\sum_{t=1}^{J^{*}}\left(R_{t}+1\right)\frac{x_{it}}{x_{i1}}-\frac{1}{J^{*}-1}.$$Hence, when:
$$\begin{array}{c} q^{*}\geq \frac{1}{n_{1}}\sum_{i=1}^{M^{*}}\sum_{t=1}^{J^{*}}\left(R_{t}+1\right)\frac{x_{it}}{x_{i1}}-\frac{1}{J^{*}-1} \end{array}$$
the equation $T_{1}\left(\eta \right)=q^{*}$ has a unique solution. □

**Theorem** **3.**
*Suppose that $X_{i1},X_{i2},\ldots ,X_{iJ^{*}}$, $i=1,\ldots ,M^{*}$ are the GPHC samples which have a density function (Equation (Equation 1)) under the censoring scheme $\left\{n_{1},T_{1},c_{1},m_{1},R_{1},R_{2},\ldots ,R_{m_{1}}\right\}$. Then, the 100(1−τ)% ECI for η is:*

$$\begin{array}{c} \left(\Psi \left({\mathrm{Data}}_{X},F_{1-\frac{\alpha }{2}}\left(2M^{*}\left(J^{*}-1\right),2M^{*}\right)\right),\Psi \left({\mathrm{Data}}_{X},F_{\frac{\alpha }{2}}\left(2M^{*}\left(J^{*}-1\right),2M^{*}\right)\right)\right) \end{array}$$

*where $0<\tau <1,{\mathrm{Data}}_{X}=c\left(X_{11},\ldots ,X_{1J^{*}},\ldots ,X_{M^{*}1},\ldots ,X_{M^{*}J^{*}}\right)$, and $\Psi \left({\mathrm{Data}}_{X},q^{*}\right)$ is the solution of η for the equation:*

$$\begin{array}{c} q^{*}=\frac{\sum_{i=1}^{M^{*}}\sum_{t=1}^{J^{*}}\left(R_{t}+1\right)\left(e^{\eta x_{it}}-1\right)}{n_{1}\left(J^{*}-1\right)\sum_{i=1}^{M^{*}}\left(e^{\eta x_{i1}}-1\right)}-\frac{1}{J^{*}-1}. \end{array}$$



**Proof** **of** **Theorem** **3.**From Equation (Equation 14), we know that:
$$\begin{array}{c} T_{1}=\frac{\sum_{i=1}^{M^{*}}\sum_{t=1}^{J^{*}}\left(R_{t}+1\right)\left(e^{\eta x_{it}}-1\right)}{n_{1}\left(J^{*}-1\right)\sum_{i=1}^{M^{*}}\left(e^{\eta x_{i1}}-1\right)}-\frac{1}{J^{*}-1} \end{array}$$
follows an F distribution with $2M^{*}\left(J^{*}-1\right)$ and $2M^{*}$ degrees of freedom. Thus, for 0<τ<1:
$$\begin{array}{cc} \; & P\left(F_{1-\frac{\tau }{2}}\left(2M^{*}\left(J^{*}\;-\;1\right),2M^{*}\right)\;<\;\frac{\sum_{i=1}^{M^{*}}\sum_{t=1}^{J^{*}}\left(R_{t}+1\right)\left(e^{\eta x_{it}}\;-\;1\right)}{n_{1}\left(J^{*}\;-\;1\right)\sum_{i=1}^{M^{*}}\left(e^{\eta x_{i1}}-1\right)}\;-\;\frac{1}{J^{*}\;-\;1}\;<\;F_{\frac{\tau }{2}}\left(2M^{*}\left(J^{*}\;-\;1\right),2M^{*}\right)\right)\\ = & P\left(\Psi \left({\mathrm{Data}}_{X},F_{1-\frac{\tau }{2}}\left(2M^{*}\left(J^{*}\;-\;1\right),2M^{*}\right)\right),\Psi \left({\mathrm{Data}}_{X},F_{\frac{\tau }{2}}\left(2M^{*}\left(J^{*}\;-\;1\right),2M^{*}\right)\right)\right)\\ = & 1-\tau \end{array}$$□

**Theorem** **4.**
*Suppose that $X_{i1},X_{i2},\ldots ,X_{iJ^{*}},\;i=1,2,\ldots ,M^{*}$ are the GPHC samples which have a density function (Equation (Equation 1)) under the censoring scheme $\left\{n_{1},T_{1},c_{1},m_{1},R_{1},R_{2},\ldots ,R_{m_{1}}\right\}$; then, the 100(1−τ)% joint ECI for η and α is:*

(17)
$$\begin{array}{c} \left\{\begin{array}{c} \Psi \left({\mathrm{Data}}_{X},F_{\tau_{1}}\left(2M^{*}\left(J^{*}\;-\;1\right),2M^{*}\right)\right)<\eta <\Psi \left({\mathrm{Data}}_{X},F_{\tau_{2}}\left(2M^{*}\left(J^{*}\;-\;1\right),2M^{*}\right)\right)\\ \frac{\eta \chi_{\tau_{1}}^{2}\left(2M^{*}J^{*}\right)}{2\sum_{i=1}^{M^{*}}\sum_{t=1}^{J^{*}}\left(R_{t}+1\right)\left(e^{\eta x_{it}}-1\right)}<\alpha <\frac{\eta \chi_{\tau_{2}}^{2}\left(2M^{*}J^{*}\right)}{2\sum_{i=1}^{M^{*}}\sum_{t=1}^{J^{*}}\left(R_{t}+1\right)\left(e^{\eta x_{it}}-1\right)} \end{array}\right. \end{array}$$

*where $\tau_{1}$ and $\tau_{2}$ are the same as Theorem 2.*


**Proof** **of** **Theorem** **4.**From Equation (Equation 15), we know that:
$$\begin{array}{c} T_{2}=2\sum_{i=1}^{M^{*}}\sum_{t=1}^{J^{*}}\left(R_{t}+1\right)\frac{\alpha }{\eta }\left(e^{\eta x_{it}}-1\right) \end{array}$$
obeys a $\chi^{2}$ distribution with $2M^{*}J^{*}$ degrees of freedom, and that it is independent of $T_{1}$. Then, for 0<τ<1:
$$\begin{array}{cc} \; & 1-\tau ={\left(\sqrt{1-\tau }\right)}^{2}\\ \; & =P\left(F_{\tau_{1}}\left(2M^{*}\left(J^{*}\;\;-\;1\right),2M^{*}\;\right)\;<\;T_{1}\;\;<\;\;F_{\tau_{2}}\left(2M^{*}\left(J^{*}\;-\;1\right),2M^{*}\right)\right)P\left(\chi_{\tau_{1}}^{2}\left(2M^{*}J^{*}\right)\;<\;T_{2}\;<\;\chi_{\tau_{2}}^{2}\left(2M^{*}J^{*}\right)\;\right)\\ \; & =P(F_{\tau_{1}}\left(2M^{*}\left(J^{*}\;-\;1\right),2M^{*}\right)\;<\;\frac{\sum_{i=1}^{M^{*}}\sum_{t=1}^{J^{*}}\left(R_{t}+1\right)\left(e^{\eta x_{it}}\;-\;1\right)}{n_{1}\left(J^{*}\;-\;1\right)\sum_{i=1}^{M^{*}}\left(e^{\eta x_{i1}}\;-\;1\right)}\;-\;\frac{1}{J^{*}\;-\;1}\;<\;F_{\tau_{2}}\left(2M^{*}\left(J^{*}\;-\;1\right),2M^{*}\;\right),\\ \; & \;\;\;\;\;\;\chi_{\tau_{1}}^{2}\left(2M^{*}J^{*}\right)<2\sum_{i=1}^{M^{*}}\sum_{t=1}^{J^{*}}\left(R_{t}+1\right)\frac{\alpha }{\eta }\left(e^{\eta x_{it}}-1\right)<\chi_{\tau_{1}}^{2}\left(2M^{*}J^{*}\right))\\ \; & =P(\Psi \left({\mathrm{Data}}_{X},F_{\tau_{1}}\left(2M^{*}\left(J^{*}\;-\;1\right),2M^{*}\right)\right)<\eta <\Psi \left({\mathrm{Data}}_{X},F_{\tau_{2}}\left(2M^{*}\left(J^{*}\;-\;1\right),2M^{*}\right)\right),\\ \; & \;\;\;\;\;\;\;\;\;\;\frac{\eta \chi_{\tau_{1}}^{2}\left(2M^{*}J^{*}\right)}{2\sum_{i=1}^{M^{*}}\sum_{t=1}^{J^{*}}\left(R_{t}+1\right)\left(e^{\eta x_{it}}-1\right)}<\alpha <\frac{\eta \chi_{\tau_{2}}^{2}\left(2M^{*}J^{*}\right)}{2\sum_{i=1}^{M^{*}}\sum_{t=1}^{J^{*}}\left(R_{t}+1\right)\left(e^{\eta x_{it}}-1\right)}) \end{array}$$□

### 3.3. Bayesian Estimation

In this section, we consider the BEs of unknown parameters from G(η,α) and Ch(θ,β) based on a GPHC sample. Assume that α,η,θ,β have independent gamma priors with the PDFs: $$\begin{array}{c} \alpha ∼\pi_{1}\left(\alpha \right)=\frac{b_{1}^{a_{1}}\alpha^{a_{1}-1}e^{-b_{1}\alpha }}{\Gamma (a_{1})};\;\;\alpha >0,a_{1}>0,b_{1}>0 \end{array}$$
$$\begin{array}{c} \eta ∼\pi_{2}\left(\eta \right)=\frac{b_{2}^{a_{2}}\eta^{a_{2}-1}e^{-b_{2}\eta }}{\Gamma (a_{2})};\;\;\eta >0,a_{2}>0,b_{2}>0 \end{array}$$
$$\begin{array}{c} \theta ∼\pi_{3}\left(\theta \right)=\frac{b_{3}^{a_{3}}\theta^{a_{3}-1}e^{-b_{3}\theta }}{\Gamma (a_{3})};\;\;\theta >0,a_{3}>0,b_{3}>0 \end{array}$$
$$\begin{array}{c} \beta ∼\pi_{4}\left(\beta \right)=\frac{b_{4}^{a_{4}}\beta^{a_{4}-1}e^{-b_{4}\beta }}{\Gamma (a_{4})};\;\;\beta >0,a_{4}>0,b_{4}>0, \end{array}$$
respectively. Then, the joint prior distribution is: $$\begin{array}{c} \omega \left(\alpha ,\eta ,\theta ,\beta \right)\propto \alpha^{a_{1}-1}\eta^{a_{2}-1}\theta^{a_{3}-1}\beta^{a_{4}-1}e^{-b_{1}\alpha -b_{2}\eta -b_{3}\theta -b_{4}\beta };\;\;\alpha ,\eta ,\theta ,\beta >0 \end{array}$$

In the statistical decision inference of Bayesian analysis, the loss function can not be ignored. Under the squared error loss function (SELF), the BE of the parameters is the posterior mean. Therefore, the BE of any function ϕ(α,η,θ,β) of α,η,θ and β under SELF is given by: (18)$$\begin{array}{c} \hat{\phi }\left(\alpha ,\eta ,\theta ,\beta \right)\;=\;E\left(\phi \left(\alpha ,\eta ,\theta ,\beta \right)|x,y\right)\;=\;\frac{\int_{0}^{\infty }\;\int_{0}^{\infty }\;\int_{0}^{\infty }\;\int_{0}^{\infty }\;\phi \left(\alpha ,\eta ,\theta ,\beta \right)\pi \left(\alpha ,\eta ,\theta ,\beta |x,y\right)d\alpha d\eta d\theta d\beta }{\int_{0}^{\infty }\;\int_{0}^{\infty }\;\int_{0}^{\infty }\;\int_{0}^{\infty }\;\pi \left(\alpha ,\eta ,\theta ,\beta |x,y\right)d\alpha d\eta d\theta d\beta } \end{array}$$

The joint posterior distribution of α,η,θ, and β is written as: (19)$$\begin{array}{ccc} \pi (\alpha ,\eta ,\theta ,\beta |x,y) & \propto \; & \omega (\alpha ,\eta ,\theta ,\beta )L(\alpha ,\eta ,\theta ,\beta |x,y)\\ & \;\;\;\propto \;\;\;\; & \prod_{i=1}^{M^{*}}\{\prod_{t=1}^{J^{*}}\alpha \theta \beta y_{i}^{\beta \;-\;1}\;\exp\left[\eta x_{it}\;-\;\frac{\alpha }{\eta }\left(e^{\eta x_{it}}\;\;-\;\;1\right)\;\left(R_{t}\;+\;\;1\right)\;-\;\frac{\alpha }{\eta }\left(e^{\eta T^{*}}\;\;\;-\;\;1\right)R_{J^{*}}^{*}\right]\\ & \;\;\;\times \;\;\;\; & \exp\left(\theta \left(1-e^{y_{i}^{\beta }}\right)\left(S_{i}+1\right)+y_{i}^{\beta }\right)\}\exp\left(\theta S_{M^{*}}^{*}\left(1-e^{{B^{*}}^{\beta }}\right)\right) \end{array}$$(20)$$\begin{array}{ccc} & \;\;\;\times \;\;\;\; & \alpha^{a_{1}-1}\eta^{a_{2}-1}\theta^{a_{3}-1}\beta^{a_{4}-1}e^{-b_{1}\alpha -b_{2}\eta -b_{3}\theta -b_{4}\beta } \end{array}$$

The conditional posterior distributions of α,η,θ, and β are given by: (21)$$\begin{array}{c} \pi_{1}\left(\alpha ,|\eta ,\theta ,\beta ,x\right)\propto \alpha^{A_{1}-1}e^{-\alpha A_{2}} \end{array}$$
where $A_{1}=M^{*}J^{*}+a_{1},A_{2}=\sum_{i=1}^{M^{*}}\sum_{t=1}^{J^{*}}\frac{1}{\eta }\left(e^{\eta x_{it}}-1\right)\left(R_{t}+1\right)+\left(\frac{1}{\eta }\left(e^{\eta T^{*}}-1\right)R_{J^{*}}\right)+b_{1}$
(22)$$\begin{array}{c} \pi_{2}\left(\eta |\alpha ,\theta ,\beta ,x\right)\propto \eta^{a_{2}-1}\;\prod_{i=1}^{M^{*}}\;\prod_{t=1}^{J^{*}}e^{\eta x_{it}}\;\exp\left[-\;\frac{\alpha }{\eta }\left(e^{\eta x_{it}}\;-\;1\right)\left(R_{t}\;+\;1\right)\right]\;\exp\left[-\;\frac{\alpha }{\eta }\left(e^{\eta T^{*}}\;\;\;-\;1\right)R_{J^{*}}\right]\;-\;b_{2}\eta \end{array}$$
(23)$$\begin{array}{c} \pi_{3}\left(\theta |\alpha ,\eta ,\beta ,y\right)\propto \theta^{A_{3}-1}e^{-\theta A_{4}} \end{array}$$
where $A_{3}=a_{3}+M^{*},A_{4}=b_{3}-\sum_{i=1}^{M*}\left(1-e^{y_{i}^{\beta }}\right)\left(S_{i}+1\right)-S_{M^{*}}^{*}\left(1-e^{{B^{*}}^{\beta }}\right)$
(24)$$\begin{array}{c} \pi_{4}\left(\beta |\alpha ,\eta ,\theta ,y\right)\propto \beta^{a_{4}+M^{*}-1}\prod_{i=1}^{M^{*}}y_{i}^{\beta -1}\exp\left(-\theta e^{y_{i}^{\beta }}\left(S_{i}+1\right)+y_{i}^{\beta }\right)\exp\left(-\theta S_{M^{*}}^{*}e^{{B^{*}}^{\beta }}-b_{4}\beta \right) \end{array}$$

It can be seen that $\pi_{1}\left(\alpha ,|\eta ,\theta ,\beta ,x,y\right)$ and $\pi_{3}\left(\theta |\alpha ,\eta ,\beta ,x,y\right)$ have gamma PDFs; thus, a sample of α and θ can be generated using the gamma distribution. However, $\pi_{2}\left(\eta |\alpha ,\theta ,\beta ,x,y\right)$ and $\pi_{4}\left(\beta |\alpha ,\eta ,\theta ,x,y\right)$ cannot be reduced to draw the sample directly by standard methods because their plots are similar to a normal distribution. We adopted the Metropolis–Hastings (MH) algorithm with a normal proposal distribution to generate η and β.

Below is a hybrid algorithm with Gibbs sampling steps for updating the parameters α and θ, and MH steps for updating η and θ. The steps of the algorithm are as follows:(1)Start with initial values of $\eta_{0}$ and $\beta_{0}$. Set t=1;(2)Generate $\alpha_{t}$ from Gamma ($A_{1}$,$A_{2}$), and $\theta_{t}$ from Gamma ($A_{3},A_{4}$);(3)Using the MH method, generate η from $\pi_{2}\left(\eta |\alpha ,\theta ,\beta ,x\right)$ with $N(\eta_{t-1},V_{\eta })$, and generate β from $\pi_{4}\left(\beta |\alpha ,\eta ,\theta ,y\right)$ with $N(\beta_{t-1},V_{\beta })$;(4)Calculate $R_{s,j}^{\left(t\right)}\left(\alpha^{\left(t\right)},\eta^{\left(t\right)},\theta^{\left(t\right)},\beta^{\left(t\right)}\right)$;(5)Repeat steps 2 to 4 *N* times, and obtain $\alpha_{t},\eta_{t},\theta_{t}$ and $\beta_{t}$, t=1,2,…,N.

The BEs of ϕ(α,η,θ,β) under the SELF are given by: $$\begin{array}{c} {\hat{\phi }}_{BS}=\frac{1}{N-M}\sum_{j=M+1}^{N}\phi_{j} \end{array}$$
where *M* is the burn-in period.

## 4. Data Analysis and Comparison Study

In this section, we show some results through numerical experiments and real data in order to compare the performance of the different methods described in the previous section.

### 4.1. Numerical Experiments

In this subsection, the performances of the MLEs and BEs under different GPHC schemes are investigated by a Monte Carlo simulation. For this purpose, the different estimates are compared in terms of bias and mean square errors (expressed as MLE-bias, BE-bias, MSE-MLE, BE-MSE, and MSE, respectively). In addition, the different confidence intervals, namely, the ACI, ECI, and HPDCI, are compared in terms of AIL and the CP. The censoring schemes are given in Table 1, and we consider s=1,j=6.

In the simulation study, we set the parameter values as follows: α=0.0061,η=0.009,β=0.2922,θ=0.009. According to Equation (Equation 7), $R_{1,6}=0.5180$. Meanwhile, according to the method of [25], GPHC samples subject to the Gompertz and Chen distributions can be generated by the following method:

**Step 1.** Generate independent and identical random variables $D_{1},D_{2},\ldots ,D_{m}$ from the standard uniform distribution U(0,1);

**Step 2.** Let $Z_{i}=-\log\left(1-D_{i}\right)$, let $Z_{1},Z_{2},\ldots ,Z_{m}$ be independent and identical, and follow the standard exponential distribution;

**Step 3.** According to the scheme $n,m,R=(R_{1},R_{2},\ldots ,R_{m})$, let: $$\begin{array}{c} Q_{1}=\frac{Z_{1}}{m} \end{array}$$
$$\begin{array}{c} Q_{i}=Q_{i-1}+\frac{Z_{i}}{\left(n-\sum_{j=1}^{i-1}R_{j}-i+1\right)};\;\;i=2,3,\ldots ,m; \end{array}$$
then, the $Q_{1},Q_{2},\ldots ,Q_{m}$ is a GPHC sample from the standard exponential distribution;

**Step 4.** Let $W_{i}=1-\exp\left(-Q_{i}\right)$; the $W_{1},W_{2},\ldots ,W_{m}$ is a GPHC sample from the standard uniform distribution;

**Step 5.** Let $X_{i:m:n}=F^{-1}\left(W_{i}\right)$, where F(x) is the CDF of the Gompertz distribution and the Chen distribution, respectively;

**Step 6.** Under the GPHC sample, there are three cases:(i)Case I: $T<X_{c:m;n}<X_{m:m:n};\;\left\{X_{1:m:n},\ldots ,X_{c:m;n}\right\}$;(ii)Case II: $X_{c:m;n}<T<X_{m:m:n};\;\left\{X_{1:m:n},\ldots ,X_{c:m;n},\ldots ,X_{J:m;n}\right\}$;(iii)Case III: $X_{c:m;n}<X_{m:m:n}<T;\;\left\{X_{1:m:n},X_{2:m;n},\ldots ,X_{m:m;n}\right\}.$

Based on the above method and the MSS model, and using the censoring scheme in Table 1 and Table 2, the observed stress sample and the corresponding strength sample are generated.

**Table 1 entropy-24-00619-t001:** Censoring scheme $n_{1}=8,n_{2}=10$.

Scheme	$m_{1}$	$T_{1}$	$c_{1}$	$m_{2}$	$T_{2}$	$c_{2}$	Censoring Scheme
I	6	500	4	5	130	3	$R=\left(0,0,\ldots ,n_{1}-m_{1}\right);S=\left(0,0,\ldots ,n_{2}-m_{2}\right)$
II							$R=\left(n_{1}-m_{1},0,\ldots ,0\right);S=\left(n_{2}-m_{2},0,\ldots ,0\right)$
III							$R=\left(0,\ldots ,0,n_{1}-m_{1},\ldots ,0\right);S=\left(0,\ldots ,0,n_{2}-m_{2},\ldots ,0\right)$
IV	5	200	3	6	500	4	$R=\left(0,0,\ldots ,n_{1}-m_{1}\right);S=\left(0,0,\ldots ,n_{2}-m_{2}\right)$
V							$R=\left(n_{1}-m_{1},0,\ldots ,0\right);S=\left(n_{2}-m_{2},0,\ldots ,0\right)$
VI							$R=\left(0,\ldots ,0,n_{1}-m_{1},\ldots ,0\right);S=\left(0,\ldots ,0,n_{2}-m_{2},\ldots ,0\right)$
VII	6	180	3	7	200	5	$R=\left(0,0,\ldots ,n_{1}-m_{1}\right);S=\left(0,0,\ldots ,n_{2}-m_{2}\right)$
VIII							$R=\left(0,0,\ldots ,n_{1}-m_{1}\right);S=\left(0,0,\ldots ,n_{2}-m_{2}\right)$
IX							$R=\left(0,\ldots ,0,n_{1}-m_{1},\ldots ,0\right);S=\left(0,\ldots ,0,n_{2}-m_{2},\ldots ,0\right)$

We obtain the BEs based on 4000 MCMC samples and discard the burn period. We repeat the process 2000 times in each scheme, and then obtain the MLEs and BEs of the parameters according to the method described in Section 3.1, Section 3.2 and Section 3.3. Finally, the BEs and MSEs of the point estimation and AILs, and the corresponding CPs of the interval estimation (95% ACI and HPDCI, ECI) for the parameters and the reliability, based on the simulation, are listed in Table 3 and Table 4.

From Table 3, we can see that the MLE-bias and BE-bias of the parameters are very small. However, the MLE has more superiority because the MSEs and the bias of the MLEs are generally lower than the BEs. For the interval estimation, it is observed that the CPs of the exact interval for the parameters and reliability are close to the HPDCI, which is better than the ACIs with respect to the AILs and CPs. Comparing Table 3 with Table 4, when $\left(n_{1},n_{2}\right)$ increases, the MSEs for the MLEs and BEs of the parameters decrease, and the AILs of the HPDCIs, ACIs, and ECIs become shorter. In terms of AILs, there is little difference between the ACI, HPDCI, and ECI in terms of parameters and reliability, but the ECIs are better than the ACIs and HPDCIs in terms of CPs.

### 4.2. Data Analysis

Here, we analyze a dataset that was first published in Musa [28] and discussed Kayal [29]. As mentioned in Section 1, in order to avoid drought in a certain area, the water storage capacity of the reservoir in any month from January to June needs to be greater than that in August of the previous year. So, we consider s=1,j=6, which suggests that it is a 1-out-of-6 system. The data (X,Y) are as follows: $$\begin{array}{c} \left(\begin{array}{cccccc} 277 & 437 & 437 & 596 & 757 & 2230\\ 277 & 363 & 405 & 522 & 535 & 613\\ 213 & 298 & 821 & 1300 & 1601 & 1620\\ 5 & 149 & 618 & 1034 & 2441 & 2640\\ 437 & 565 & 714 & 927 & 1119 & 4462 \end{array}\right)\left(\begin{array}{c} 135\\ 340\\ 277\\ 874\\ 460 \end{array}\right) \end{array}$$We first verify that the Chen distribution and the Gompertz distribution can be fitted to the given dataset. Based on the method described in Section 3.1, Section 3.2 and Section 3.3, we obtain the MLEs of unknown parameters and compare them with the other two lifetime distributions, including inverse Weibull and exponential distributions. In Table 5, we record the Kolmogorov–Smirnov (K-S) statistics along with their corresponding p-values. From this table, we observe that, compared with other distributions, the Chen distribution and the Gompertz distribution provide a pretty good fitting for a given dataset.

Scheme 1: $T_{1}=1350;n_{1}=6,m_{1}=4,c_{1}=2,R=\left(1,0,0,1\right);T_{2}=200;n_{2}=5,$$m_{2}=3,c_{2}=2,S=\left(0,2,0\right)$.Scheme 2: $T_{1}=1700;n_{1}=6,m_{1}=5,c_{1}=2,R=\left(1,0,0,0,0\right);T_{2}=300;n_{2}=5,$$m_{2}=4,c_{2}=3,S=\left(1,0,0,0\right)$.

To obtain the censoring sample from *X* and *Y*, we first censor some elements from the *Y* by the method explained in Section 2. For any censoring data of *Y*, we remove the same row of the *X* sample. In the remaining sample of *X*, we apply the censoring scheme for each row. Comparing Table 6 with Table 7, we can see that the MLEs and BEs of the parameters are very close to the complete sample with respect to bias and MSE. However, the BE is not affected by the initial value; it has more advantages, and the MSE of the BE is generally less than the MLE.

For interval estimation, it is observed that the CPs of the exact interval and HPDCI for the parameters are close. Comparing scheme I with scheme II, $T_{1}$ and $T_{2}$ in the scheme have a great effect. In scheme II, due to the large setting of $T_{1}$ and $T_{2}$, the results are closer to complete samples. The HPDCIs are better than the ACIs in terms of the AILs and CPs, the MSEs for the MLE and the BEs of the parameters decrease, and the AILs of the HPDCIs and ACIs become shorter.

## 5. Conclusions

Industrial safety accidents occur frequently, mainly because the accuracy of reliability estimations of industrial system equipment are unable to meet the specific requirements. Therefore, it is of great significance to select appropriate estimation methods, progressive censoring life tests, and estimation evaluation criteria to improve the accuracy of system reliability estimation in existing industrial systems. The present study utilized the extensively used multicomponent system stress–strength model (also referred to as the *s*-out-of-*j* system) and the GPHC scheme, along with two point-estimation methods, namely, maximum likelihood and Bayesian methods, and three interval-estimation methods, namely, ACI, HPDCI, and ECI. Repeated numerical simulation tests were performed, which, together with a set of reservoir storage data, revealed that the GPHC scheme could ensure the accuracy of the reliability estimation of the MSS model. Moreover, limiting the distribution types of the stress and strength variables was not required in the setting of the model, and a further accurate reliability estimation value of the model could be obtained within a short life test duration and at a reduced cost. In addition, numerical experiments were conducted, the results of which indicated that the CPs of the ECI for the parameters were close to the HPDCI, and the HPDCI was better than the ACIs of the MSS model parameters and reliability, in terms of the AILs and CPs. In terms of AIL, there was little difference among the ACI, HPD, and ECI of the parameters, although the ECIs were better than the ACIs and HPDCIs in terms of CPs. In addition, $T_{1}$ and $T_{2}$ in the scheme greatly affected the reliability of the MSS model. In practice, setting $T_{2}$ greater than $T_{1}$ would enable the reliability estimation GPHC sample to be closer to that under the complete sample, which would reduce costs.

The GPHC scheme is similar to the progressive type-II censoring scheme in a special case. This scheme is a generalization of the progressive and hybrid censoring schemes, due to which the results could be further extended. On the other hand, as the distribution types of the model variables were not limited in the present study, the reliability of the MSS model has complex forms, and there is no relevant mathematical and statistical theory support; thusm the ECI of the MSS model reliability could not be obtained. In the present study, the multicomponent system was limited to a non-repairable system, while industrial production often involves repairable multicomponent systems. Therefore, further research could involve the combination of the multicomponent repairable stress–strength model and a censoring scheme.

## Figures and Tables

**Table 2 entropy-24-00619-t002:** Censoring scheme $n_{1}=8,n_{2}=10$.

Scheme	$m_{1}$	$T_{1}$	$c_{1}$	$m_{2}$	$T_{2}$	$c_{2}$	Censoring Scheme
I	12	500	8	15	130	13	$R=\left(0,0,\ldots ,n_{1}-m_{1}\right);S=\left(0,0,\ldots ,n_{2}-m_{2}\right)$
II							$R=\left(n_{1}-m_{1},0,\ldots ,0\right);S=\left(n_{2}-m_{2},0,\ldots ,0\right)$
III							$R=\left(0,\ldots ,0,n_{1}-m_{1},\ldots ,0\right);S=\left(0,\ldots ,0,n_{2}-m_{2},\ldots ,0\right)$
IV	10	500	8	18	130	15	$R=\left(0,0,\ldots ,n_{1}-m_{1}\right);S=\left(0,0,\ldots ,n_{2}-m_{2}\right)$
V							$R=\left(n_{1}-m_{1},0,\ldots ,0\right);S=\left(n_{2}-m_{2},0,\ldots ,0\right)$
VI							$R=\left(0,\ldots ,0,n_{1}-m_{1},\ldots ,0\right);S=\left(0,\ldots ,0,n_{2}-m_{2},\ldots ,0\right)$
VII	9	200	8	18	200	15	$R=\left(0,0,\ldots ,n_{1}-m_{1}\right);S=\left(0,0,\ldots ,n_{2}-m_{2}\right)$
VIII							$R=\left(0,0,\ldots ,n_{1}-m_{1}\right);S=\left(0,0,\ldots ,n_{2}-m_{2}\right)$
IX							$R=\left(0,\ldots ,0,n_{1}-m_{1},\ldots ,0\right);S=\left(0,\ldots ,0,n_{2}-m_{2},\ldots ,0\right)$

**Table 3 entropy-24-00619-t003:** Bias, MSEs of MLE and BE, and length and CP of ACI, HPD, and ECI when $n_{1}=10,n_{2}=8$.

Scheme	Parameter and Reliability	MLE-Bias	MLE-MSE	ACI-AIL	ACI-CP	BE-Bias	BE-MSE	HPDCI-AIL	HPDCI-CP	ECI-AIL	ECI-CP
I	α	0.00207	6.34 × 10^−6^	0.00827	0.793	0.0083	8.44 × 10^−5^	0.009932	0.69	5.32 × 10^−4^	0.82
	η	0.00582	2.07 × 10^−5^	0.02306	0.903	0.00889	7.91 × 10^−7^	1.92 × 10^−4^	0.817	1.86 × 10^−4^	0.709
	β	0.0957	0.0134	0.2791	0.744	0.0548	0.00438	0.2458	0.98	0.0245	0.83
	θ	0.0089	1.49 × 10^−4^	0.0302	0.759	0.01553	0.00476	0.0702	0.96	0.00897	0.93
	$R_{1,6}$	5.51 × 10^−6^	3.32 × 10^−7^	1.7784	0.869	0.192	4.58 × 10^−4^	0.521	0.76		
II	α	0.0112	4.182 × 10^−6^	0.00399	0.757	0.00987	7.969 × 10^−5^	1.476 × 10^−3^	0.57	1.43 × 10^−3^	0.67
	η	0.00536	1.001 × 10^−5^	0.0106	0.508	0.00748	8.25 × 10^−4^	0.432	0.83	0.00113	0.62
	β	0.05498	0.00871	0.1123	0.577	0.105	0.0927	0.107	0.9	0.0856	0.767
	θ	0.00421	9.463 × 10^−5^	0.01098	0.327	0.0347	0.00769	0.5947	0.86	0.01123	0.97
	$R_{1,6}$	6.7 × 10^−6^	4.54 × 10^−6^	0.766	0.722	0.1835	8.37 × 10^−4^	0.0176	0.84		
III	α	0.00256	8.89 × 10^−6^	0.00698	0.622	0.0103	8.49 × 10^−5^	0.0824	0.79	0.00476	0.675
	η	0.01323	2.78 × 10^−5^	0.0235	0.737	0.00765	7.761 × 10^−5^	1.946 × 10^−4^	0.85	0.00123	0.61
	β	0.11708	0.0186	0.2549	0.571	0.0572	0.00849	0.3794	0.58	0.2067	0.77
	θ	0.00844	1.53 × 10^−4^	0.0204	0.364	0.0984	0.00498	0.69443	0.84	0.0215	0.404
	$R_{1,6}$	3.27 × 10^−7^	4.85 × 10${}^{-8}$	1.384	0.717	0.198	9.54 × 10^−4^	0.541	0.93		
IV	α	0.00107	3.107 × 10^−6^	0.00311	0.667	0.0107	8.56 × 10^−7^	0.00465	0.65	0.00577	0.9
	η	0.00332	1.22 × 10^−5^	0.0109	0.828	0.197	8.49 × 10^−7^	0.06701	0.75	0.10972	0.819
	β	0.0335	0.00358	0.114	0.872	0.497	4.37 × 10^−5^	0.097	0.74	0.0991	0.67
	θ	0.00418	7.50 × 10^−5^	0.0156	0.536	0.00976	3.21 × 10^−4^	0.0843	0.87	0.0985	0.92
	$R_{1,6}$	7.92 × 10^−5^	3.67 × 10${}^{-6}$	0.727	0.922	0.0859	2.85 × 10^−3^	0.0946	0.89		
V	α	1.14 × 10^−3^	3.66 × 10^−6^	3.23 × 10^−3^	0.64	0.274	8.95 × 10^−4^	0.9647	0.87	0.0143	0.77
	η	0.00789	1.21 × 10^−5^	0.0107	0.585	0.967	3.47 × 10^−6^	0.529	0.695	0.0198	0.533
	β	0.05081	0.00682	0.1008	0.531	0.08421	8.79 × 10^−4^	0.0796	0.59	0.0935	0.67
	θ	0.00471	1.003 × 10^−4^	0.01315	0.741	0.354	0.00627	0.894	0.48	0.156	0.69
	$R_{1,6}$	4.55 × 10^−5^	3.62 × 10^−6^	0.789	0.856	0.265	0.00439	0.0845	0.86		
VI	α	0.00245	4.77 × 10^−6^	0.00785	0.607	0.367	7.45 × 10^−7^	0,854	0.82	0.00977	0.66
	η	0.00789	1.211 × 10^−5^	0.0107	0.528	0.967	3.475 × 10^−6^	0.106	0.73	0.00989	0.576
	β	0.0512	0.0085	0.1186	0.674	0.095	4.31 × 10^−4^	0.0894	0.573	0.1187	0.56
	θ	0.00613	1.443 × 10^−4^	0.01142	0.776	0.385	0.00747	0.886	0.94	0.00986	0.89
	$R_{1,6}$	8.76 × 10^−7^	5.21 × 10^−8^	0.887	0.857	0.134	0.00578	0.0845	0.85		
VII	α	8.13 × 10^−4^	1.91 × 10^−6^	0.00304	0.806	0.0085	4.77 × 10^−4^	0.00875	0.61	0.00478	0.872
	η	0.00214	8.92 × 10^−6^	0.00810	0.89	0.00475	2.89 × 10^−5^	0.00875	0.61	0.00478	0.872
	β	0.03012	0.00283	0.0998	0.83	0.10927	0.0543	0.1734	0.87	0.0854	0.91
	θ	0.00429	1.077 × 10${}^{-3}$	0.0185	0.66	0.0747	0.00654	0.604	0.88	0.1374	0.87
	$R_{1,6}$	6.77 × 10${}^{-6}$	2.31 × 10${}^{-7}$	0.8223	0.994	0.253	4.36 × 10${}^{-4}$	0.176	0.845		
VIII	α	8.97 × 10${}^{-4}$	12.39 × 10${}^{-6}$	0.003145	0.776	0.00625	2.89 × 10${}^{-5}$	0.00875	0.675	0.00978	0.94
	η	0.00510	8.79 × 10${}^{-6}$	0.00803	0.83	0.0784	8.27 × 10${}^{-4}$	0.875	0.89	0.102	0.85
	β	0.0436	0.005118	0.0916	0.618	0.145	0.157	0.365	0.989	0.189	0.779
	θ	0.00457	9.15 × 10${}^{-5}$	0.0163	0.646	0.0214	0.00645	0.604	0.878	0.189	0.929
	$R_{1,6}$	5.84 × 10${}^{-5}$	2.37 × 10${}^{-6}$	0.9298	0.982	0.2431	5.36 × 10${}^{-5}$	-.7894	0.889		
IX	α	0.00104	2.86 × 10${}^{-6}$	2.61 × 10${}^{-3}$	0.604	0.00857	4.32 × 10${}^{-5}$	8.78 × 10${}^{-3}$	0.72	4.57 × 10${}^{-3}$	0.92
	η	0.00571	1.21 × 10${}^{-5}$	0.00825	0.822	0.0949	3.19 × 10${}^{-5}$	0.275	0.67	0.118	0.92
	β	0.0489	0.00633	0.0928	0.678	0.195	0.0957	0.365	0.929	0.177	0.827
	θ	0.00406	6.28 × 10${}^{-5}$	0.0123	0.0846	0.00214	0.0645	0.619	0.817	0.0189	0.819
	$R_{1,6}$	5.32 × 10${}^{-6}$	2.09 × 10${}^{-6}$	0.788	0.915	0.946	0.188	7.89 × 10${}^{-3}$	0.0889		

**Table 4 entropy-24-00619-t004:** Bias, MSEs of MLE and BE, and AIL and CP of ACI, HPD, and ECI when $n_{1}=15,n_{2}=20$.

Scheme	Parameter and Reliability	MLE-Bias	MLE-MSE	ACI-AIL	ACI-CP	BE-Bias	BE-MSE	HPDCI-AIL	HPDCI-CP	ECI-AIL	ECI-CP
I	α	7.957 × 10${}^{-4}$	9.41 × 10${}^{-7}$	0.00345	0.892	7.95 × 10${}^{-5}$	2.95 × 10${}^{-5}$	0.003369	0.74	0.00276	0.891
	η	1.769 × 10${}^{-3}$	1.167 × 10${}^{-5}$	0.00783	0.918	9.41 × 10${}^{-4}$	3.87 × 10${}^{-5}$	6.77 × 10${}^{-5}$	0.58	0.000798	0.898
	β	0.0343	1.94 × 10${}^{-3}$	0.136	0.881	8.19 × 10${}^{-3}$	4.67 × 10${}^{-5}$	0.1132	0.92	0.009962	0.918
	θ	0.00621	7.196 × 10${}^{-5}$	0.0271	0.733	1.19 × 10${}^{-4}$	2.89 × 10${}^{-6}$	0.0339	0.92	0.02798	0.787
	$R_{1,6}$	5.41 × 10${}^{-7}$	5.33 × 10${}^{-6}$	0.9182	0.992	0.0222	9.26 × 10${}^{-3}$	0.3079	0.68		
II	α	9.05 × 10${}^{-3}$	1.244 × 10${}^{-6}$	0.00356	0.84	2.49 × 10${}^{-4}$	4.88 × 10${}^{-5}$	0.030396	0.79	0.00458	0.91
	η	0.00786	1.151 × 10${}^{-5}$	0.00801	0.527	7.71 × 10${}^{-4}$	8.37 × 10${}^{-5}$	8.57 × 10${}^{-5}$	0.52	0.00708	0.928
	β	0.03605	0.00207	0.1303	0.849	4.56 × 10${}^{-3}$	7.27 × 10${}^{-5}$	0.09986	0.9	0.01062	0.91
	θ	0.00563	4.767 × 10${}^{-5}$	0.0224	0.689	1.08 × 10${}^{-4}$	3.57 × 10${}^{-6}$	0.0993	0.9	0.08789	0.792
	$R_{1,6}$	2.34 × 10${}^{-7}$	1.87 × 10${}^{-6}$	0.8384	0.923	0.0860	7.98 × 10${}^{-3}$	0.5079	0.76		
III	α	7.32 × 10${}^{-4}$	8.141 × 10${}^{-7}$	0.00312	0.887	2.05 × 10${}^{-5}$	3.37 × 10${}^{-5}$	0.009336	0.78	0.00376	0.981
	η	1.31 × 10${}^{-3}$	1.12 × 10${}^{-5}$	0.00571	0.919	2.07 × 10${}^{-5}$	4.96 × 10${}^{-6}$	5.89 × 10${}^{-4}$	0.67	0.00719	0.98
	β	0.0298	0.00141	0.1174	0.885	8.19 × 10${}^{-3}$	4.67 × 10${}^{-5}$	0.1132	0.92	0.009962	0.918
	θ	0.00534	4.75 × 10${}^{-5}$	0.0236	0.747	2.42 × 10${}^{-5}$	6.77 × 10${}^{-6}$	0.0457	0.98	0.037796	0.87
	$R_{1,6}$	1.79 × 10${}^{-7}$	9.17 × 10${}^{-6}$	0.819	0.929	0.0879	1.34 × 10${}^{-3}$	0.2522	0.65		
IV	α	0.00105	1.59 × 10${}^{-6}$	0.00331	0.74	8.79 × 10${}^{-4}$	1.58 × 10${}^{-4}$	0.00874	0.74	0.00302	0.815
	η	0.002726	1.47 × 10${}^{-5}$	0.00951	0.836	9.41 × 10${}^{-4}$	3.87 × 10${}^{-5}$	6.77 × 10${}^{-5}$	0.58	0.000798	0.898
	β	0.0274	0.00121	0.1187	0.919	6.69 × 10${}^{-3}$	5.21 × 10${}^{-5}$	0.1067	0.98	0.01099	0.908
	θ	5.64 × 10${}^{-3}$	5.96 × 10${}^{-5}$	0.0271	0.8	8.15 × 10${}^{-4}$	7.98 × 10${}^{-6}$	0.0839	0.97	0.003589	0.87
	$R_{1,6}$	7.19 × 10${}^{-6}$	9.87 × 10${}^{-7}$	0.836	0.927	0.0892	7.56 × 10${}^{-3}$	0.2097	0.728		
V	α	9.72 × 10${}^{-4}$	1.335 × 10${}^{-6}$	0.0031	0.761	9.85 × 10${}^{-4}$	2.55 × 10${}^{-4}$	0.008369	0.77	0.00306	0.898
	η	0.00161	1.418 × 10${}^{-5}$	0.00562	0.842	7.74 × 10${}^{-5}$	6.37 × 10${}^{-5}$	2.17 × 10${}^{-5}$	0.42	0.000598	0.792
	β	0.0261	0.00112	0.1121	0.908	2.89 × 10${}^{-4}$	5.76 × 10${}^{-5}$	0.1032	0.91	0.1099	0.918
	θ	0.00536	4.88 × 10${}^{-5}$	0.0257	0.794	1.19 × 10${}^{-4}$	2.89 × 10${}^{-6}$	0.0339	0.92	0.02798	0.787
	$R_{1,6}$	7.72 × 10${}^{-5}$	9.17 × 10${}^{-6}$	0.805	0.921	0.0222	9.26 × 10${}^{-3}$	0.3079	0.68		
VI	α	1.31 × 10${}^{-4}$	1.74 × 10${}^{-7}$	0.0003904	0.744	8.92 × 10${}^{-5}$	1.36 × 10${}^{-5}$	0.003062	0.77	0.00376	0.802
	η	2.36 × 10${}^{-4}$	1.96 × 10${}^{-6}$	7.944 × 10${}^{-4}$	0.864	3.24 × 10${}^{-5}$	5.77 × 10${}^{-5}$	2.62 × 10${}^{-5}$	0.62	0.000298	0.898
	β	0.00376	1.701 × 10${}^{-4}$	0.01567	0.883	2.09 × 10${}^{-4}$	3.57 × 10${}^{-5}$	0.0924	0.9	0.01098	0.898
	θ	7.29 × 10${}^{-3}$	6.71 × 10${}^{-6}$	0.00354	0.773	3.52 × 10${}^{-3}$	2.99 × 10${}^{-6}$	0.0339	0.92	0.00298	0.827
	$R_{1,6}$	2.76 × 10${}^{-4}$	5.23 × 10${}^{-6}$	0.1121	0.9854	1.43 × 10${}^{-4}$	9.26 × 10${}^{-5}$	0.2085	0.68		
VII	α	0.00129	2.24 × 10${}^{-6}$	0.00326	0.646	1.82 × 10${}^{-3}$	2.95 × 10${}^{-5}$	0.003069	0.64	0.00376	0.891
	η	0.00359	1.75 × 10${}^{-5}$	0.0108	0.788	2.41 × 10${}^{-3}$	3.87 × 10${}^{-5}$	6.07 × 10${}^{-5}$	0.682	0.00798	0.778
	β	0.0272	0.001185	0.1189	0.913	8.52 × 10${}^{-2}$	4.67 × 10${}^{-3}$	0.1587	0.92	0.09678	0.918
	θ	0.00538	5.19 × 10${}^{-5}$	0.0253	0.786	1.19 × 10${}^{-3}$	7.72 × 10${}^{-5}$	0.0579	0.767	0.10253	0.987
	$R_{1,6}$	4.71 × 10${}^{-6}$	1.21 × 10${}^{-7}$	0.8244	0.914	2.02 × 10${}^{-7}$	8.46 × 10${}^{-7}$	0.1064	0.69		
VIII	α	0.00155	2.975 × 10${}^{-6}$	0.00295	0.756	8.054 × 10${}^{-3}$	1.74 × 10${}^{-6}$	0.00797	0.62	0.00278	0.878
	η	0.00461	2.095 × 10${}^{-5}$	0.00683	0.832	0.00475	2.89 × 10${}^{-6}$	0.00875	0.813	0.00478	0.882
	β	0.0252	0.00102	0.1117	0.92	0.10927	0.0544	0.1734	0.83	0.0927	0.91
	θ	0.00532	5.144 × 10${}^{-5}$	0.0261	0.807	0.00489	1.297 × 10${}^{-3}$	0.0485	0.69	0.00747	0.854
	$R_{1,6}$	5.64 × 10${}^{-5}$	1.31 × 10${}^{-6}$	0.8004	0.908	0.0860	6.24 × 10${}^{-5}$	0.9279	0.9604		
IX	α	6.27 × 10${}^{-4}$	1.12 × 10${}^{-6}$	1.25 × 10${}^{-3}$	0.604	8.77 × 10${}^{-4}$	1.56 × 10${}^{-5}$	9.98 × 10${}^{-3}$	0.71	4.57 × 10${}^{-3}$	0.92
	η	0.00193	8.39 × 10${}^{-6}$	0.00308	0.792	0.0877	2.31 × 10${}^{-6}$	0.0275	0.74	0.00118	0.857
	β	0.0115	0.00489	0.0507	0.915	0.195	0.0957	0.278	0.959	0.197	0.998
	θ	0.00252	2.94 × 10${}^{-5}$	0.0118	0.801	0.00845	0.0645	0.627	0.717	0.0189	0.819
	$R_{1,6}$	7.91 × 10${}^{-5}$	1.65 × 10${}^{-6}$	0.358	0.975	9.46 × 10${}^{-5}$	1.88 × 10${}^{-6}$	7.89 × 10${}^{-3}$	0.0889		

**Table 5 entropy-24-00619-t005:** K-S distances and *p*-values for parameters and reliability, based on different distributions.

Distribution	Data X				Distribution	Data Y			
	α	η	K-S	*p*-Value		β	θ	K-S	*p*-Value
Gompertz	0.0010	1.960 × 10${}^{-5}$	0.1502	0.5077	Chen	0.2922	0.0021	0.2130	0.9389
IW	0.5705	25.8900	0.2516	0.04	IW	1.7360	15157	0.2181	0.90
Exponential		0.0010	0.1536	0.15	Exponential		0.0024	0.2852	0.70

**Table 6 entropy-24-00619-t006:** MLEs, BEs, AILs, and HPDs for parameters based on the complete data.

Parameter and Reliability	MLE	ACI	BE	HPDCI
α	0.0010	(4.34 × 10${}^{-4}$, 1.63 × 10${}^{-3}$)	0.00096	(6.40 × 10${}^{-4}$, 1.31 × 10${}^{-3}$)
η	1.96 × 10${}^{-5}$	(6.40 × 10${}^{-4}$, 1.31 × 10${}^{-3}$)	0.00014	(1.41 × 10${}^{-4}$, 1.47 × 10${}^{-4}$)
β	0.2922	(0.221, 0.362)	0.2956	(0.2916, 0.2928)
θ	0.0021	(−4.712 × 10${}^{-3}$, 9.10 × 10${}^{-3}$)	0.00484	(0.00185, 0.00253)
$R_{1,6}$	0.991	(0.7709942, 1.211296)	0.9760	(0.9046621, 1.0003050)

**Table 7 entropy-24-00619-t007:** MLEs, BEs, AILs, and HPDCIs for parameters based on the censoring data.

Scheme	Parameter and Reliability	MLE	ACI	BE	HPDCI	ECI
	α	5.55 × 10${}^{-4}$	(−0.189, 0.190)	0.00107	(2.41 × 10${}^{-4}$, 1.67 × 10${}^{-3}$)	(6.42 × 10${}^{-6}$, 3.50 × 10${}^{-5}$)
	η	6.87 × 10${}^{-4}$	(−6.03 × 10${}^{-4}$, 1.96 × 10${}^{-3}$)	0.00087	(6.08 × 10${}^{-5}$, 2.02 × 10${}^{-3}$)	(−5.6 × 10${}^{-3}$, 5.08 × 10${}^{-3}$)
I	β	0.3742	(0.271, 1.356)	0.258132	(0.054, 0.424)	(0.217, 0.493)
	θ	0.000142	(−0.0032, 0.0033)	0.0216	(2.98 × 10${}^{-8}$, 1.29 × 10${}^{-1}$)	(4.88 × 10${}^{-3}$, 1.33 × 10${}^{-7}$)
	$R_{1,6}$	0.9837	(0.542, 1.424)	0.9878	(0.347, 0.998)	
	α	7.08 × 10${}^{-4}$	(−7.36 × 10${}^{-4}$, 2.15 × 10${}^{-3}$)	2.73 × 10${}^{-4}$	(−3.24 × 10${}^{-4}$, 1.25 × 10${}^{-3}$)	(5.77 × 10${}^{-6}$, 3.14 × 10${}^{-5}$)
	η	1.08 × 10${}^{-3}$	(−1.47 × 10${}^{-3}$, 3.63 × 10${}^{-3}$)	3.46 × 10${}^{-4}$	(−1.07 × 10${}^{-3}$, 1.96 × 10${}^{-3}$)	(−3.83 × 10${}^{-4}$, 6.09 × 10${}^{-3}$)
II	β	0.3851	(−0.1663, 0.937)	0.0921	(0.0017, 0.3448)	(0.162, 0.495)
	θ	7.69 × 10${}^{-5}$	(−0.0020, 0.0022)	7.43 × 10${}^{-3}$	(−1.42 × 10${}^{-3}$, 3.05 × 10${}^{-2}$)	(1.82 × 10${}^{-7}$, 4.05 × 10${}^{-3}$)
	$R_{1,6}$	0.985	(0.378, 1.592)	0.993	(0.364, 0.994)	

## Data Availability

The data are available from the corresponding author upon request.

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
