# Peer review of "Multicomponent Stress–Strength Model Based on Generalized Progressive Hybrid Censoring Scheme: A Statistical Analysis"

_entropy, 2022, doi:10.3390/e24050619_

Round 1

Reviewer 1 Report

Review of Multicomponent stress-strength model based on generalized progressive hybrid censoring scheme: a statistical analysis

In the present study, the authors propose a statistical analysis of the reliability of the multicomponent stress-strength model under a generalized progressive hybrid censoring scheme. The distribution of the stress and strength are taken to follow respectively the Chen and Gompertz distributions. Thus, after a large formal development, they estimate the reliability by optimizing its parametrization through a Newton-Raphson method. Different statistical estimations on the reliability and its parameters are performed and compared. 

General notes on the article.

Many standard rules of punctuation are not respected: space after the comma, uppercase after points (or points before uppercase). In many sentences, verbs are missing or misused words are misspelled. 

This report proposes several corrections in this sense but the authors should consider them as a small subset of all the improvements needed. Thus, some crucial definitions are difficult to understand and hinder comprehension of the rest of the article.

Nevertheless, the content of the study looks promising. If the problems of writing could be solved, this article could become an interesting study worth publishing. As a suggestion, the writer can use an assisting program (e.g. Grammarly, or check here: https://www.mdpi.com/authors/english) that would avoid most mistakes and improve his style.

Because of its low quality of writing and is not suitable for publication in its current state, a substantial revision is needed.

Abstract

lines 1-3: Punctuation should be corrected

line 3: generalized progressive hybrid censoring scheme(GPHC) -> generalized progressive hybrid censoring (GPHC) scheme

line 10: “..., we have not obtained an analytical solution” -> not necessary to say it, go directly to the estimations. 

line 12: the HPDCI acronym is not consistent with the words before it (however clearly defined in the list of abbreviations).

It would be helpful to fully rewrite the abbreviations since the abstract. Anyway, if the author chose not to do it, MSS should not be redefined in line 39.

1 Introduction

Line 79: “as follows” must be followed by “:” not by a point. I suggest simply removing “The rest of this paper is organized as follows.

2 Model

  • Line 85 (style): Introduction of Gompertz distribution clumsy, please reformulate.
  • Line 88: “parameters,and” -> add space after coma.
  • Line 93: The expression simplifies to exponential power distribution.
  • Line 95: “which cite” -> where is cited
  • Line 96: I Could not find the publication of Tian [25], how could this article be reached?
  • In lines 96 to 104, important definitions for the comprehension of the rest of the article are mentioned but poorly formulated. Please rewrite it. Besides,
  • Line 98: The sum of R_i ends with “+m=n”, this does not look right 
  • A space should be added between “CASE” and the number
  • Line 105: “In Case III, it is the progressive II censoring scheme” -> “Case III corresponds to the progressive II censoring scheme”.
  • Line 110: “In the s-out-of-j: G system, it consists” -> clumsy, please improve
  • Line 115: The Ch function appears for the first time here, it looks it corresponds to the Chen distribution defined in lines 90-94? Please clarify.
  • Line 121: “first we” -> “we first”, or propose a better formulation.

3 Estimation of parameters and reliability

  • Line 128: Same as before, the Ch function has not been defined
  • Line 133: I suggest using the definition symbol instead of “=” since the x_it and y_it are alternative symbols of previously defined variables.
  • Line 135: the “complex” word can be confusing, do you refer as “complex number” or “complicated” (I guess the second), please clarify.
  • Line 143-144: Please reformulate
  • Line 146: The N4 function is not defined or mentioned in the text.
  • Line 150: “is certain value” -> “is some value”
  • Line 158: “identically distribution“ -> “Identically distributed
  • Lines 208-210: Please correct the formulation of Theorem 3, not questioning the math, only the language, “support”-> “suppose”, punctuations…
  • Line 221: obeys a chi…
  • Line 225: please reformulate
  • Line 233: “function(SELF)” -> “function (SELF)” (add space) 
  • Line 236: is written as (add “is”)
  • Please write on separate lines the definition of A1, A2, A3, and A4.

  1. Data analysis and comparison study

In all this section the orthography “Gomgertz” must be corrected to “Gompertz”

  • Line 268: Clickable to reference [29] is broken
  • Line 288: MCMC not defined, does that stand for “Monte Carlo”?
  • Line 324: in respect of -> with respect to,
  • Lines 324-325, 326-328: please reformulate

The conclusions should be developed further, including final conclusions. Possible prospects for future studies or improvements would also be welcome.

References:

Please add space after “,” and “;”. 

Reviewer 2 Report

Given my background, I can only judge the part concerning statistical inference which, although correct, does not present any kind of novelty. The notation is very heavy and difficult to follow. Furthermore, for a non-expert reader, the introduction to the problem is insufficient and the justification of the relevance of the proposed results is missing. There are many typos and spacing errors. I don't think the article deserves to be published on Entropy

Round 2

Reviewer 1 Report

Even if there has been a noticeable effort to improve this manuscript, it still requires work. In particular, the English language and style do not match the level of articles published in MPDI. Some important definitions have not been clearly presented and make the rest of the manuscript hard to follow. Many mistakes still need to be corrected, Grammarly still finds some but I believe that most of them must be found by humans, in particular, in sentences that involve mathematical symbols. I strongly encourage the authors to submit for review by a native speaker in English.

I still believe that the ideas, method, and results presented in this article are worth publishing, but not in the current state. 

The content of the introduction has significantly improved, examples of multi-component systems are given which make it easier to understand the ideas used later. Nevertheless, the language needs to be improved.

Why is there an additional section in the article? Putting a discussion after the conclusion is odd, this should be revised, I suggest simply merging the discussion paragraph in the conclusion. However, I appreciate its content which provides a look closer at reality when it comes to evaluating multi-component systems.

I would suggest removing any reference that is not accessible to any reader. 

Introduction:

line 9 : Add space between interval and (ACI) 

line 26: formal terms‘stress’and ‘strength’ -> add corresponding spaces

line 36: Indeed,with advancements in science and technology ->add space

can be improved with somthing like “With the progress of science and technology”

line 39: Therefore, investigating the reliability of multicomponent systems in the stress–strength (MSS) model. -> Incomplete sentence, missing a verb.

line 40: The system comprised j components and is subjected to a stress. -> No transition is with the previous sentence, this sentence is really clumsy, and must be improved (e.g. The system has j components and is constrained.)

line 42: ”Such a system functions when …”-> confusing, change “functions” with “works” 

line 48-50, please connect the examples, it is very hard to follow. Is the example of solar panels related to “storage capacity of the reservoir” (what reservoir?)?

line 55: Rao [8–10],on the other -> add space after comma.

line 72: obeyed -> obey

line 80: is meaningful work -> is a meaningful work

line 87-89: Please reformulate

line 92: Specifically,the -> add space after comma.

line 101: and the real data analysis -> remove “the”

2 Model

line 108: what is denote by G(eta,alpha), the sentence is not clear (eta and alpha should be exchanged no?)

line 114: same construction of sentence as the last point, what does the Ch function refer to? Is it the same as F?

line 116: mention Singh -> add space

line 119: and its life is assumed  -> I guess it is the life of each unit, the sentence should be corrected.

line 119: The sentence “Before the life test…” is still not well written. In addition: are c and m chosen randomly? It is not explained.

line 122: “follows: The” -> “the” (lowercase)

lines 122-125: These sentences still need to be rewritten, there are missing spaces and verbs. This section is important for the rest of the paper, please explain it carefully.

between 132 and 133: “CaseXX :” -> “Case XX:” (correct spaces)

3 Estimation of parameters and reliability

line 142: “Likelihood Estimation(MLE)” -> add space

line 145-147. Please reformulate, the sentence it difficult to read with so many mathematical characters. For instance, T* could be explained with words (in addition to its formal definition).

line 155: I suppose that “t” is an index of x and y (like in the likelihood function written above), so it should be written as an index (lower).

line 158: I suppose you mean the MLEs of alpha, eta, theta, and beta without “^”, I would understand that the “^” would be assigned to the values obtained for these variables after applying the Newton-Raphson method.

line 158: “is that” -> replace by “is:”

line 169: “dalta method” -> what is the dalta method?

line 175: “with their” -> “with their corresponding values”

line 182: “C obey a Chi^2(2)” -> “C obeys”, missing the “distribution” word somewhere. What does the “(2)” stands for?

line 182: “D has follows” -> please reformulate

line 262: “PDF,thus,sample” -> please add space after commas.

line 265: “Metropolis-Hastings(MH)” -> add space before “(”

  1. Data analysis and comparison study

line 284: “error(express as …” -> error (expressed as … 

line 288-290: please reformulate

lines 291 and 293: “independent and identically” -> “independent and identical” (twice) 

line 295: “),Let” -> please add space

line 303-305: “Case XX” -> “Case XX:” (add two points)

line 317: “Are generally less” -> “are generally lower”

line 330,335, caption of table 5: “Gomgerz” -> “Gompertz”

line 331: “Based on the above method in section 3.1, 3.2, 3.3” -> “Based on the method described in sections 3.1, 3.2, and 3.3”

line 332: “and compared” -> “and compare”

line 333: “, We” -> “, we”

line 343: “To obtain the censoring sample from X and Y, we perform as follows. First, we censor” -> “To obtain the censoring sample from X and Y, we first censor”

Conclusion

line 357: “frequently occur” -> “occur frequently”

line 375: It would be great to recall here what represents T1 and T2 e.g. “time of …” (I am not expecting an additional sentence, this should be clearly explained when they appear first)

Reviewer 2 Report

The authors have clarified in the new version of the Introduction and Discussion the explanation of the problem and its relevance in the systems reliability context. They also have fixed typos and spacing errors. Since my concerns weren't about the statistical part, I think the paper has been sufficiently improved to warrant publication in Entropy.
